# A Multi-Level Framework for Accelerating Training Transformer Models

**Longwei Zou, Han Zhang, Yangdong Deng**
Tsinghua University
{zoulw22,han-zhan20}@mails.tsinghua.edu.cn dengyd@tsinghua.edu.cn

## Abstract

The fast growing capabilities of large-scale deep learning models, such as Bert, GPT and ViT, are revolutionizing the landscape of NLP, CV and many other domains. Training such models, however, poses an unprecedented demand for computing power, which incurs exponentially increasing energy cost and carbon dioxide emissions. It is thus critical to develop efficient training solutions to reduce the training costs. Motivated by a set of key observations of inter- and intra-layer similarities among feature maps and attentions that can be identified from typical training processes, we propose a multi-level framework for training acceleration. Specifically, the framework is based on three basic operators, Coalescing, De-coalescing and Interpolation, which can be orchestrated to build a multi-level training framework. The framework consists of a V-cycle training process, which progressively down- and up-scales the model size and projects the parameters between adjacent levels of models via coalescing and de-coalescing. The key idea is that a smaller model that can be trained for fast convergence and the trained parameters provides high-qualities intermediate solutions for the next level larger network. The interpolation operator is designed to break the symmetry of neurons incurred by de-coalescing for better convergence performance. Our experiments on transformer-based language models (e.g. Bert, GPT) as well as a vision model (e.g. DeiT) prove that the proposed framework reduces the computational cost by about 20% on training BERT/GPT-Base models and up to 51.6% on training the BERT-Large model while preserving the performance. [1]

## 1 Introduction

Recent years have witnessed an unprecedented success of large scale deep learning models in areas such as natural language processing (NLP), computer vision (CV) and graph analysis. BERT (Devlin et al., 2019), GPT (Radford et al., 2018; 2019; Brown et al., 2020; OpenAI, 2023) and other large language models have revolutionized the landscape of NLP, while the transformer based CV models (e.g. ViT model (Dosovitskiy et al., 2021)) and more conventional convolutional neural networks are demonstrating remarkable performance in vision processing applications. The excellent performance of these large models, however, demands an ever rising amount of computing power increasing exponentially with the model size and thus escalating energy cost and carbon emissions. For example, LLaMA-65B was trained with 2048 A100 GPUs in a period of 21 days(Touvron et al., 2023) and took an estimated cost of $4 million [2]. Moreover, it has become a growing concern of the increasing monopoly of large models by a few giant companies who can afford the excessive developing cost. It is thus pressing to seek efficient solutions to accelerating the training process for large models so as to democratize the large AI models to a wider audience.

Increasing the number of parameters in deep learning models proves to be the most effective way to improve the expressiveness of deep neural networks and incur emergent capabilities. However, the training complexity is also largely determined by the number of parameters in the pervasively used gradient-based descent formulation in which the parameters are incrementally adjusted to minimize

---

[1] Code is available at https://github.com/Photooon/Multi-Level-Training-Framework

[2] The estimation is based on an average cost of $3.93 per A100 GPU per hour on Google Cloud Platform: https://cloud.google.com/.

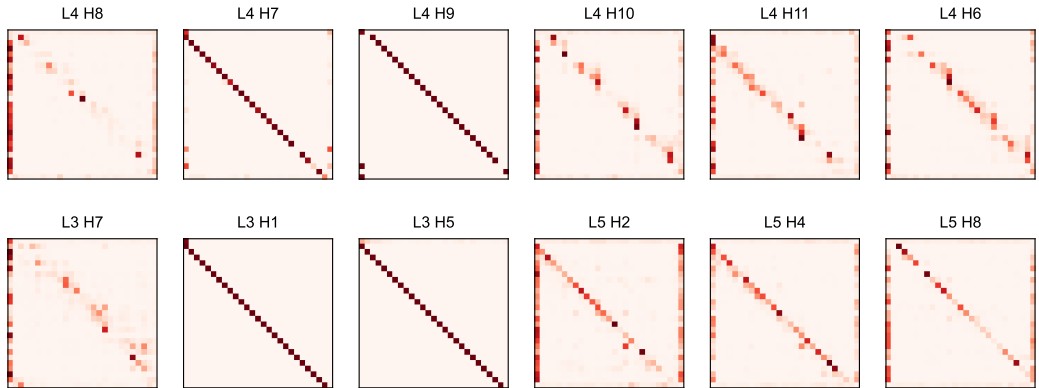

Figure 1: Visualization of attention patterns in BERT-Base of a randomly chosen sample sentence. The darker the color, the more attention a token pays to another one. The first row shows the attention patterns of various heads on layer 4 and represents the similarity within a layer. The second row shows the attention patterns for adjacent layers of layer 4, i.e., layer 3 and layer 5, and demonstrates the similarity between layers. The similarities inter and intra-layers offer the potential for accelerating training with multi-level framework.

the difference between the model output and the ground truth result. Therefore, it is appealing to develop a training framework that works on a reduced set of parameters but at the same time reserve expressiveness of the underlying model. We realized that the well-known multigrid algorithm (Trottenberg et al., 2000), which is an efficient numerical procedure to solve linear equations and partial differential equations, actually offers such a potential. The central idea is to re-structure the underlying problem into a hierarchy of interrelated grids. Solving the problem on the coarse grid is equivalent to removing low frequency errors. The resultant solution process can be more efficient as only a reduced number of variables have to be tackled. The solution is then mapped back to the fine grid as an initialization for the removal of high frequency errors. In comparison with traditional solvers, multigrid enables remarkable performance in terms of convergence speed and robustness (Stuben, 2001). Inspired by the central computing pattern of the multigrid algorithm, we perform a systematic investigation on accelerating the training of large models with a multi-level framework.

The feasibility of a multi-level training framework is justified by the following three observations. First, in a given family of models, the smaller ones (e.g. BERT-Base and GPT-Base) always converge faster than the bigger ones (e.g. BERT-Large and GPT-Large) but deliver a lower level of expressiveness. Second, our experimental results on BERT demonstrates that there are many similar patterns in each layer. Figure 1 visualizes the attention patterns of a randomly selected sentence on BERT-Base. The attention patterns of various attention heads extracted from layer 4 are shown in the first row. It can be seen that many heads, e.g. the 7th head (L4 H7) and 9th head (L4 H9), have almost identical patterns. The similarity within a layer has also been discovered in Convolutional Neural Networks (Zeiler & Fergus, 2014). Third, previous works show that a given model tends to exhibit significant similarities among adjacent layers. In (Gong et al., 2019), the authors indicate that neighboring layers in a transformer model have similar attention patterns. We illustrates some attention patterns from layer 3 and layer 5 at the second row in Figure 1. The results confirm the observation of (Gong et al., 2019). In addition to the above observations, a few recent works (Gong et al., 2019; Yang et al., 2020; Chen et al., 2016; 2022) exploit the fast convergence of a smaller model to train a large scale model by progressively increasing the model size. Such works can be regarded as special cases of our multi-level framework with a single de-coalescing operation that maps the parameters of a small network to a larger one.

In this work, we propose the first overall framework for multi-level training of deep learning models. The framework is built upon three key operators, Coalescing, De-coalescing, and Interpolation. Consisting a V-cycle working flow, our multi-level framework progressively down- and up-scales the model size and project the parameters between a models in adjacent levels. The fast convergence of smaller models provides high-quality intermediate solutions for the next level larger network, which saves computational cost. Figure 2 demonstrates the V-cycle training process with 2 levels.

The major contributions of this work are as follows. (1) We propose an efficient multi-level training framework for transformer based deep learning models. (2) We develop a formal formulation for the three basic operators and introduce guidelines to design these operators for numerical robustness

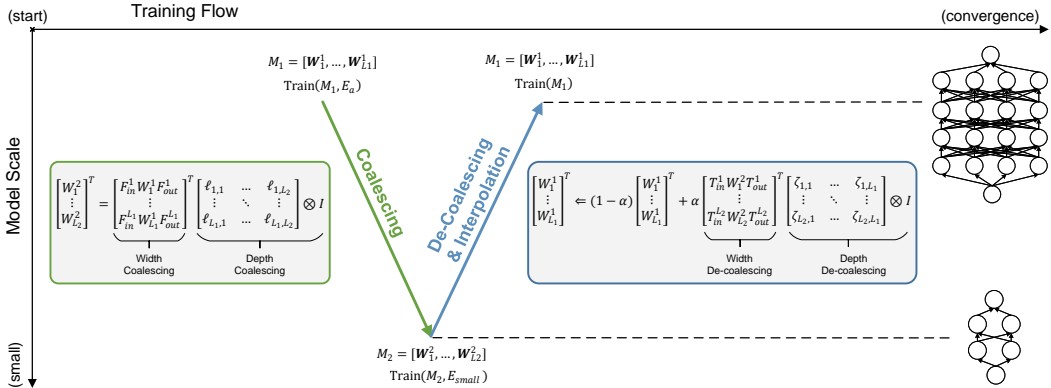

Figure 2: A 2-level V-cycle training process. $M_1$ with parameters of $[\boldsymbol{W}_1^1, ..., \boldsymbol{W}_{L_1}^1]$ is the original model to train. $M_2$ with parameters of $[\boldsymbol{W}_1^2, ..., \boldsymbol{W}_{L_2}^2]$ is a smaller model which is coalesced from $M_1$ by coalescing intra- and inter-layer neighboring nodes. $\boldsymbol{F}_{in}^l$ and $\boldsymbol{F}_{out}^l$ are used to decrease the input and output dimension of parameter $\boldsymbol{W}_l^1$ in layer $l$. The depth coalescing matrix is decomposed into an $L_1 \times L_2$ matrix and an identity matrix via Kronecker product. We first train the $M_1$ model for $E_a$ epochs to initialize model parameters. Then we train the coalesced $M_2$ model, which converges faster. After that, we de-coalesce the parameters of $M_2$ to the original size and interpolate them with the parameters of $M_1$ before coalescing. Finally, we continue to train the interpolated $M_1$ model.

and training performance. (3) We conduct extensive experiments on two transformer-based language model, BERT and GPT, as well as a vision model, DeiT. The results show that the proposed framework enables considerable speed-up in training large scale models. Compared with the traditional training methods, our multi-level framework reduces the training cost by 20% on BERT-Base and GPT-Base, 51.6% on Bert-Large, and 27.1% on DeiT-B.

## 2    RELATED WORK

### 2.1    EFFICIENT PRE-TRAINING

Various efficient pre-training techniques have been developed. Most of these, e.g., ELECTRA (Clark et al., 2020), large batch optimization (You et al., 2020), layer dropping (Zhang & He, 2020), token dropping (Hou et al., 2022), and weight sharing (Yang et al., 2021), are orthogonal to our work. KI (Qin et al., 2022) distills knowledge from small models into larger models for efficient pre-training. A few recent works (Wen et al., 2020; Haber et al., 2018; Chang et al., 2018; Gaedke-Merzhäuser et al., 2020; Cyr et al., 2019; Gong et al., 2019; Yang et al., 2020) pre-train a large scale model by progressively stacking layers. Meanwhile, Net2Net (Chen et al., 2016) increases network complexity in the width direction by copying neurons and also expands depth by adding identity layers to preserve the learned features. bert2BERT (Chen et al., 2022) extends Net2Net to the transformer architecture. In Network Expansion (Ding et al., 2023), the authors propose to impose the orthogonality across the expanded filters in CNN and utilize the exponential moving averaged model to expand the ViT along the depth direction. In CompoundGrow (Gu et al., 2021), the authors prove it is beneficial to expand the network in width, depth, and tokens together. In the staged training approach (Shen et al., 2022), Shen et al. focus on growth operators scheduling to keep the training dynamics. LiGO (Wang et al., 2023) learns a linear mapping matrix between the small and the large model through stochastic gradient descent (SGD). Similar approaches (Wu et al., 2020; Wang et al., 2021) are also proposed to accelerate the training of other architectures. Most of the above approaches start from a small model and progressively increase the model size by reusing parameters. Therefore, they should be regarded as special cases of the multi-level framework with only de-coalescing operation.

### 2.2    MULTIGRID

Our work is inspired by the multigrid method (Trottenberg et al., 2000), which is an efficient numerical procedure to solve linear equations and partial differential equations with multiple levels. The method has received successful applications in many application domains such as finite element

method (Zhu & Cangellaris, 2006), eigenvalue computation (Knyazev & Neymeyr, 2003), and graph partitioning (Abou-Rjeili & Karypis, 2006). Of course, a direct porting of the multigrid method to neural network training, which is under a different set of constraints, is not feasible. Borrowing the spirit of the multigrid solvers, we propose a systematic investigation on accelerating the training of large models with a multi-level framework.

## 3   METHODOLOGY

We aim to accelerate the training of large transformer models by leveraging the fast convergence behaviors of a smaller model derived from the larger ones through a coarsening operation. We propose three operators, Coalescing, De-coalescing, and Interpolation to develop a complete working flow. These operators merge parameters, transform the parameters back to the original model, and correct the parameters of the original model with de-coalescing parameters, respectively. The design of the operators enable the framework to reduce most errors in a small model at a less total training cost. In the following sections, we first introduce the Coalescing, De-coalescing and Interpolation operators and then elaborate the V-cycle training process built upon them.

For simplicity, we assume that all layers are feed forward layers without bias and have the same input and output dimensions. We give the generalization for other components of transformer in Appendix A. We use $M_k, k = 1, 2, ..K$ to refer to the model at level $k$. $M_1$ is the original model and the level number $k$ increases by one when we coarsen the model to a smaller one. We use $\boldsymbol{W}_l^k \in \mathbb{R}^{d_{in}^k, d_{out}^k}$ to represent the parameter of layer $l$ in model $M_k$. $d_{in}^k$ and $d_{out}^k$ refer to the input and output dimensions. $\text{sum}_{\text{row/col}}(\boldsymbol{A})$ denotes the summation of each row or column of matrix $\boldsymbol{A}$, resulting a sum vector. We adopt $\text{diag}(\boldsymbol{a})$ to convert a vector $\boldsymbol{a}$ into a diagonal matrix.

### 3.1   COALESCING

As the first step, we coalesce the model in the width direction, while keeping the layered structure untouched. Then we coalesce the model in the depth direction to get a smaller model coarsened in both width and depth directions.

**Width Coalescing**   $\boldsymbol{W}_l^k \in \mathbb{R}^{d_{in}^k, d_{out}^k}$ is the parameter of layer $l$ in the model $M_k$. We transform it into $\boldsymbol{U}_l^{k+1} \in \mathbb{R}^{d_{in}^{k+1}, d_{out}^{k+1}}$ with width coalescing matrices $\boldsymbol{F}_{in}^{k+1,l} \in \mathbb{R}^{d_{in}^{k+1}, d_{in}^k}$ and $\boldsymbol{F}_{out}^{k+1,l} \in \mathbb{R}^{d_{out}^k, d_{out}^{k+1}}$ for in and out dimensions as follows.

$$\boldsymbol{U}_l^{k+1} = \boldsymbol{F}_{in}^{k+1,l} \boldsymbol{W}_l^k \boldsymbol{F}_{out}^{k+1,l} \tag{1}$$

To ensure the output of layer $l$ is properly received by layer $l + 1$ and stabilize the output of each layer, we define $\boldsymbol{F}_{in}^{k+1,l+1}$ as:

$$\boldsymbol{F}_{in}^{k+1,l+1} = \boldsymbol{F}_{out}^{k+1,l T} \text{diag}(1/\text{sum}_{\text{col}}(\boldsymbol{F}_{out}^{k+1,l} \boldsymbol{F}_{out}^{k+1,l T})) \tag{2}$$

where $\text{diag}(1/\text{sum}_{\text{col}}(\boldsymbol{F}_{out}^{k+1,l} \boldsymbol{F}_{out}^{k+1,l T}))$ is used to normalize the product of $\boldsymbol{F}_{out}^{k+1,l}$ and $\boldsymbol{F}_{out}^{k+1,l T}$. The width coalescing matrix $\boldsymbol{F}_{out}^{k+1,l}$ is arbitrary as long as the matrix has full column rank. The matrix we used is detailed in Section 4.1.

**Depth Coalescing**   After width coalescing, we have $\boldsymbol{U}_l^{k+1}, l = 1, 2, ..., L_k$, and need to transpose them into $\boldsymbol{W}_l^{k+1}, l = 1, 2, ..., L_{k+1}$. $L_k$ and $L_{k+1}$ are layer numbers of the model $M_k$ and $M_{k+1}$. $L_{k+1} < L_k$ always holds. To reduce the depth, we use a depth coalescing matrix $\boldsymbol{R}^{k+1} \in \mathbb{R}^{L_k, L_{k+1}, d_{in}^{k+1}, d_{out}^{k+1}}$ to map those parameters as follows.

$$[\boldsymbol{W}_1^{k+1}, \boldsymbol{W}_2^{k+1}, \ldots, \boldsymbol{W}_{L_{k+1}}^{k+1}] = [\boldsymbol{U}_1^{k+1}, \boldsymbol{U}_2^{k+1}, \ldots, \boldsymbol{U}_{L_k}^{k+1}] \boldsymbol{R}^{k+1} \tag{3}$$

As we can see, the dimension of $\boldsymbol{R}^{k+1}$ can be large. Therefore, we decompose it into an $L_k \times L_{k+1}$ matrix with elements of $\ell_{i,j}^{k+1}, i = 1, ..., L_k, j = 1..., L_{k+1}$ and an identity matrix via Kronecker product as in LiGO(Wang et al., 2023):

$$\boldsymbol{R}^{k+1} = \begin{bmatrix} \ell_{1,1}^{k+1} & \cdots & \ell_{1,L_{k+1}}^{k+1} \\ \vdots & \ddots & \vdots \\ \ell_{L_k,1}^{k+1} & \cdots & \ell_{L_k,L_{k+1}}^{k+1} \end{bmatrix} \otimes \boldsymbol{I} \tag{4}$$

where the $\otimes$ is the Kronecker product that multiplies $\ell_{i,j}^{k+1}$ with the identity matrix respectively.

In general, we coalesce the parameters of $M_k$ to those of a smaller model $M_{k+1}$ as follows.

$$\begin{bmatrix} \boldsymbol{W}_1^{k+1} \\ \vdots \\ \boldsymbol{W}_{L_{k+1}}^{k+1} \end{bmatrix}^T = \begin{bmatrix} \boldsymbol{F}_{in}^{k+1,1}\boldsymbol{W}_1^k\boldsymbol{F}_{out}^{k+1,1} \\ \vdots \\ \boldsymbol{F}_{in}^{k+1,L_k}\boldsymbol{W}_{L_k}^k\boldsymbol{F}_{out}^{k+1,L_k} \end{bmatrix}^T \begin{bmatrix} \ell_{1,1}^{k+1} & \cdots & \ell_{1,L_{k+1}}^{k+1} \\ \vdots & \ddots & \vdots \\ \ell_{L_k,1}^{k+1} & \cdots & \ell_{L_k,L_{k+1}}^{k+1} \end{bmatrix} \otimes \boldsymbol{I} \tag{5}$$

## 3.2 DE-COALESCING

De-coalescing is the inverse operator of Coalescing. To map the parameters back to the original model, we first do depth de-coalescing on model $M_{k+1}$ to increase the depth, and then perform width de-coalescing to increase the width based on the result of depth de-coalescing.

**Depth De-coalescing**  Given a model $M_{k+1}$, which has parameters $\boldsymbol{W}_l^{k+1} \in \mathbb{R}^{d_{in}^{k+1}, d_{out}^{k+1}}, l = 1, 2, ..., L_{k+1}$. We could use the depth de-coalescing matrix $\boldsymbol{G}^{k+1} \in \mathbb{R}^{L_{k+1}, L_k, d_{in}^k, d_{in}^k}$ to transform them into $\boldsymbol{U}_l^{k+1} \in \mathbb{R}^{d_{in}^{k+1}, d_{out}^{k+1}}, l = 1, 2, ..., L_k$.

$$[\boldsymbol{U}_1^{k+1}, \boldsymbol{U}_2^{k+1}, ..., \boldsymbol{U}_{L_k}^{k+1}] = [\boldsymbol{W}_1^{k+1}, \boldsymbol{W}_2^{k+1}, ..., \boldsymbol{W}_{L_{k+1}}^{k+1}]\boldsymbol{G}^{k+1} \tag{6}$$

As in the previous section, we decompose the depth de-coalescing matrix $\boldsymbol{G}$ into an $L_{k+1} \times L_k$ matrix with elements of $\zeta_{i,j}, i = 1, ..., L_{k+1}, j = 1, ..., L_k$ and an identity matrix $\boldsymbol{I}$.

$$\boldsymbol{G}^{k+1} = \begin{bmatrix} \zeta_{1,1}^{k+1} & \cdots & \zeta_{1,L_k}^{k+1} \\ \vdots & \ddots & \vdots \\ \zeta_{L_{k+1},1}^{k+1} & \cdots & \zeta_{L_{k+1},L_k}^{k+1} \end{bmatrix} \otimes \boldsymbol{I} \tag{7}$$

Consider that we are doing the depth de-coalescing on the result of depth coalescing as follows.

$$[\boldsymbol{U}_1^{k+1}, \boldsymbol{U}_2^{k+1}, ..., \boldsymbol{U}_{L_k}^{k+1}] = [\boldsymbol{U}_1^{k+1}, \boldsymbol{U}_2^{k+1}, ..., \boldsymbol{U}_{L_k}^{k+1}]\boldsymbol{R}^{k+1}\boldsymbol{G}^{k+1} \tag{8}$$

To keep the value of parameters in each layer stable after coalescing and de-coalescing i.e., the column sum of $\boldsymbol{R}^{k+1}\boldsymbol{G}^{k+1}$ equals to $\boldsymbol{I}$, we have the $\boldsymbol{G}^{k+1}$ as:

$$\boldsymbol{G}^{k+1} = \boldsymbol{R}^{k+1^T}\text{diag}(1/\text{sum}_{\text{col}}(\boldsymbol{R}^{k+1}\boldsymbol{R}^{k+1^T})) \tag{9}$$

where the $\text{diag}(1/\text{sum}_{\text{col}}(\boldsymbol{R}^{k+1}\boldsymbol{R}^{k+1^T}))$ performs a normalization of the parameters.

**Width De-coalescing**  After depth de-coalescing, we have $\boldsymbol{U}_l^{k+1} \in \mathbb{R}^{d_{in}^{k+1}, d_{out}^{k+1}}$. We use width de-coalescing matrices $\boldsymbol{T}_{in}^{k+1,l}$ and $\boldsymbol{T}_{out}^{k+1,l}$ to transform them into $\boldsymbol{W}_l^k \in \mathbb{R}^{d_{in}^k, d_{out}^k}$ for in and out dimensions, respectively.

Recalling the width coalescing, we have the following equation after coalescing and de-coalescing.

$$
\begin{aligned}
\boldsymbol{W}_l^k &= \boldsymbol{T}_{in}^{k+1,l} \boldsymbol{U}_l^{k+1} \boldsymbol{T}_{out}^{k+1,l} \\
&= \boldsymbol{T}_{in}^{k+1,l} \boldsymbol{F}_{in}^{k+1,l} \boldsymbol{W}_l^k \boldsymbol{F}_{out}^{k+1,l} \boldsymbol{T}_{out}^{k+1,l}
\end{aligned}
\tag{10}
$$

Similar to the definition of depth de-coalescing, we define $\boldsymbol{T}_{in}^{k+1,l}$ and $\boldsymbol{T}_{out}^{k+1,l}$ as follows.

$$
\begin{aligned}
\boldsymbol{T}_{in}^{k+1,l} &= \operatorname{diag}(1/\operatorname{sum}_{\text{row}}(\boldsymbol{F}_{in}^{k+1,l^T}\boldsymbol{F}_{in}^{k+1,l}))\boldsymbol{F}_{in}^{k+1,l^T} \\
\boldsymbol{T}_{out}^{k+1,l} &= \boldsymbol{F}_{out}^{k+1,l^T}\operatorname{diag}(1/\operatorname{sum}_{\text{col}}(\boldsymbol{F}_{out}^{k+1,l}\boldsymbol{F}_{out}^{k+1,l^T}))
\end{aligned}
\tag{11}
$$

In summary, we de-coalesce the model $M_{k+1}$ to a larger model $M_k$ as follows.

$$
\begin{bmatrix} \boldsymbol{W}_1^k \\ \vdots \\ \boldsymbol{W}_{L_k}^k \end{bmatrix}^T
=
\begin{bmatrix} \boldsymbol{T}_{in}^{k+1,1}\boldsymbol{W}_1^{k+1}\boldsymbol{T}_{out}^{k+1,1} \\ \vdots \\ \boldsymbol{T}_{in}^{k+1,L_{k+1}}\boldsymbol{W}_{L_{k+1}}^{k+1}\boldsymbol{T}_{out}^{k+1,L_{k+1}} \end{bmatrix}^T
\begin{bmatrix} \zeta_{1,1}^{k+1} & \cdots & \zeta_{1,L_k}^{k+1} \\ \vdots & \ddots & \vdots \\ \zeta_{L_{k+1},1}^{k+1} & \cdots & \zeta_{L_{k+1},L_k}^{k+1} \end{bmatrix} \otimes \boldsymbol{I}
\tag{12}
$$

## 3.3 INTERPOLATION

Since we define the de-coalescing matrix as the normalized transposition of the coalescing matrix, it's easy to find that the parameter $\boldsymbol{W}_l^k$ obtained after de-coalescing is far from full rank. In the worst case, half of the elements of $\boldsymbol{W}_l^k$ will be identical to the respective ones in the other part (Chen et al., 2016). Such a value distribution significantly limits the capability of the model. We call it the symmetry of neurons.

Instead of injecting noise (Chen et al., 2016) or parameters from higher layers (Chen et al., 2022) to alleviate this problem, we propose a more general operator, interpolation, which (1) transfers knowledge back to the larger model $M_k$ from the de-coalesced smaller model, (2) avoids similar adjacent parameters incurred by de-coalescing, and (3) improves the convergence rate further. We find that the interpolation operation allows the acceleration scales with the times of model enlargement. On the contrary, a higher number of mapping in the previous literature leads to a worsened convergence speed. This empirical observation is elaborated in Appendix B.

Specifically, after training the smaller model $M_{k+1}$ coalesced from $M_k$, we first de-coalesce $M_{k+1}$ to the size of $M_k$ and get model $M_{k,de-coalesced}$. Then we merge the $M_k$ and $M_{k,de-coalesced}$ under the control of a hyperparameter $\alpha \in (0,1)$.

$$
M_k \Leftarrow (1-\alpha)M_k + \alpha M_{k,de-coalesced}
\tag{13}
$$

In addition to alleviate the problem of symmetric neurons, $\alpha$ also determines how much updated knowledge is incorporated into a larger model from a smaller one. The effect of $\alpha$ is discussed in Appendix D. In most cases, $\alpha = 0.25$ suffices to give satisfying results.

## 3.4 V-CYCLE TRAINING PROCESS

With the Coalescing, De-coalescing, and Interpolation operators, we can build a V-cycle training process as inpired by the V-cycle in multigrid method (Trottenberg et al., 2000). We leave other training processes like W-cycle and FMG in multigrid method for future work.

Algorithm 1 expounds the V-cycle training process. At first, it progressively coalesces the model to reduce the model complexity and then trains the smallest model for $E_{small}$ epochs. The hyperparameter $E_{small}$ is used to allow early stop of training smaller models at the end of the fast convergence phase. Next, the smaller model is de-coalesced and interpolated to set the parameters of a larger model. Finally, the model size reaches its original size and $M_1$ will be further trained until convergence. In addition, we train the models at each level for $E_a$ epochs before coalescing for parameters initialization. $E_a$ is set to the number of warm-up steps. We always stop the training of smaller models for one half of the number of steps for the large model. We further discuss the robustness of hyper-parameters in Appendix D

---

**Algorithm 1:** V-cycle Training Process

---

   **Input** : untrained model $M_1$, dataset $D$, number of levels $K$, interpolation factor $\alpha$, epochs
            $E_{small}$ for training smaller models, epochs $E_a$ for initializing.
   **Output:** the pre-trained model $M_1$

1 **for** $l = 1 \rightarrow K - 1$ **do**
2    |   update $M_l$ on $D$ for $E_a$ epochs
3    |   $M_{l+1}$ = Coalescing($M_l$) // decrease model size, more details in Algorithm 2
4 **end**
5 **for** $l = K \rightarrow 2$ **do**
6    |   update $M_l$ for $E_{small}$ epochs
7    |   $M_{l-1,de-coalesced}$ = De-coalescing($M_l$) // recover size, more details in Algorithm 3
8    |   $M_{l-1}$ = Interpolation($M_{l-1}$, $M_{l-1,de-coalesced}$, $\alpha$) // more details in Algorithm 4
9 **end**
10 update $M_1$ on $D$ until convergence

---

## 4 EXPERIMENT

### 4.1 EXPERIMENTAL SETUP

**Architectures and Datasets** We exercise our multi-level framework on two language models, BERT (Devlin et al., 2019) and GPT (Radford et al., 2018), as well as a vision model, DeiT (Touvron et al., 2021). We use English Wikipedia and BooksCorpus (Zhu et al., 2015) as pre-training data for BERT and GPT, while DeiT is trained with ImageNet (Deng et al., 2009). For evaluation, we test the pre-trained BERT on the GLUE benchmark (Wang et al., 2019). We evaluate the pre-trained GPT on LAMBADA (Paperno et al., 2016), PTB, WikiText-2 and WikiText103 under a zero-shot setting without fine-tuning on the training set. CIFAR10 (Krizhevsky et al., 2009), CIFAR100 (Krizhevsky et al., 2009), Flowers102 (Nilsback & Zisserman, 2008), and Stanford-Cars (Krause et al., 2013) are adopted to test the downstream performance of DeiT.

**Baselines** We evaluate our framework by comparing it with five baselines. The first baseline is a the model trained from scratch. The second is StackBERT (Gong et al., 2019), in which the depth is increased by progressively stacking layers. [3] The third baseline is bert2BERT (Chen et al., 2022) that extends the width with advanced knowledge initialization. The fourth one is LiGO (Wang et al., 2023) that learns to expand the network both in depth and width. The fifth method is Network ExpansionDing et al. (2023) that expands the model with the exponential moving averaged parameters. We choose KI (Qin et al., 2022) that distills knowledge from a small model into a larger model as the fifth baseline. Specifically, bert2BERT, LiGO, and KI all assume that there are well-trained small models and do not consider the training cost of smaller models. To be fair, we take into account the training cost of smaller models for bert2BERT, LiGO and KI when comparing with them.

**Implementation Details** During pre-training, we train the BERT-Base with the following settings, 40 training epochs, 10K warm-up steps, a peak learning rate of 1e-4, and a batch size of 512. We remove the next sentence prediction task (Liu et al., 2019) and use a fixed sequence length of 128. We use the same settings for BERT-Large. In the case of GPT-Base, we use 20 training epochs, 10K warm-up steps, a peak learning rate of 1e-4, and a batch size of 256. We train the DeiT-B with a peak learning rate of 1e-3, 300 training epochs and a batch size of 1024.

For the evaluation of BERT-Base and BERT-Large, we take the tasks from the GLUE benchmark. We choose a batch size of 32, a learning rate from {5e-6, 1e-5, 2e-5, 3e-5, 5e-5}, and train the model with 5 epochs on all GLUE fine-tuning tasks. We run each training process for three times with random seeds for GLUE. We evaluate the GPT-Base on LAMBADA (Paperno et al., 2016), PTB, WikiText-2, and WikiText103 under zero-shot setting, i.e., without fine-tuning. For DeiT-B, we fine-tune the pre-trained model for 1000 epochs with a batch size of 768 and a learning rate of 0.01 under SGD optimizer and the same data augmentation in the training.

---

[3]We implemented a MSLT(Yang et al., 2020) based training procedure, which is similar to StackBERT. We find that it's hard for MSLT to reach sufficient performance when trained from scratch as observed by bert2BERT (Chen et al., 2022). So we only include the results of StackBERT.

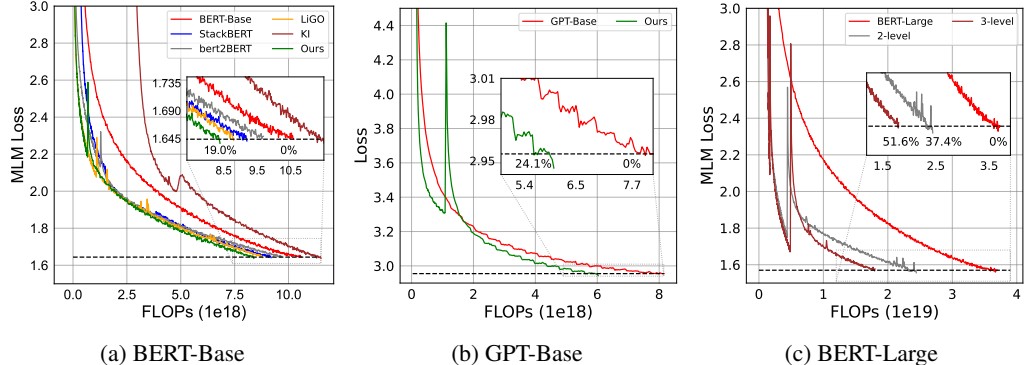

| (a) BERT-Base | (b) GPT-Base | (c) BERT-Large |

Figure 3: Results on BERT-Base, GPT-Base and BERT-Large. (a-c) show loss curves of BERT-Base, GPT-Base and BERT-Large pre-training. The dashed lines are the final results of models training from scratch. For BERT-Base and GPT-Base, our approach saving about 20% computational costs. For BERT-Large, we save 37.4% training cost with 2-level training process and 51.6% with 3-level.

In the multi-level training, we coalesce the model to reduce width and depth by half. We define the width coalescing matrix as $\boldsymbol{F}_{w,out}^{k+1,l} = [\boldsymbol{I}/2, \boldsymbol{I}/2]^T$ and elements of depth coalescing matrix as $\ell_{2i-1,i}^{k+1} = \ell_{2i,i}^{k+1} = 0.5, i = 1, ..., L_{k+1}$. Intuitively, they coalesce two neighboring neurons and merge two adjacent layers. The de-coalescing matrices can be derived from Eq. 9 and Eq. 11. We use $\alpha = 0.25$ for GPT and DeiT, $\alpha = 0.5$ for BERT. Unless otherwise noted, we train the model with a two-level mapping. The length of initializing stages for BERT/GPT and DeiT are 10K steps and five epochs. We always stop the training of the smaller model halfway through the training. For example, the BERT training steps are 300K and thus $E_{small}$ will be 150K steps.

Our experiments are conducted on NVIDIA A100 GPUs. As the mixed precision training (Micikevicius et al., 2018) and DeepSpeed framework (Rajbhandari et al., 2020) are orthogonal to our method, we use both for the pre-training of BERT and GPT.

## 4.2 EXPERIMENTAL RESULTS

Table 1: GLUE benchmark results between our approach and baselines. BERT-Base means the model is trained from scratch. All results on GLUE benchmark are run in three times with different seeds and numbers in parentheses are the standard deviation across the runs.

| Method | Saving (FLOPs) | Saving (Walltime) | SST-2 (Acc) | MNLI (Acc) | MRPC (Acc) | CoLA (Mcc) | QNLI (Acc) | QQP (Acc) | STS-B (Acc) | Avg |
|---|---|---|---|---|---|---|---|---|---|---|
| BERT-Base | 0% | 0% | 89.6(0.5) | 77.8(0.2) | 78.7(0.2) | 52.9(0.3) | 85.1(0.5) | 89.4(0.1) | 83.5(0.4) | 79.7(0.2) |
| StackBERT | 15.2% | 8.2% | 89.6(0.4) | 77.5(0.2) | 71.7(0.4) | 52.1(0.5) | 85.8(0.3) | 89.3(0.1) | 82.3(0.4) | 79.3(0.1) |
| bert2BERT | 8.9% | -2.4% | 89.5(0.5) | 77.7(0.1) | 76.5(0.9) | 49.5(0.4) | 84.8(0.4) | 88.3(0.1) | 81.0(0.4) | 78.2(0.3) |
| LiGO | 17.4% | 7.9% | 89.2(0.3) | 79.1(0.2) | 77.4(2.1) | 53.7(0.3) | 85.9(0.1) | 89.0(0.1) | 83.6(0.3) | 79.7(0.3) |
| Network Expansion | 14.8% | 8.1% | 89.6(0.3) | 77.6(0.2) | 76.1(0.6) | 53.4(0.4) | 85.5(0.3) | 89.6(0.1) | 82.7(0.4) | 79.4(0.1) |
| KI | -6.9% | -25.9% | 89.9(0.2) | 78.3(0.2) | 76.7(0.4) | 55.1(0.6) | 85.6(0.3) | 89.6(0.1) | 84.7(0.3) | 79.9(0.1) |
| Ours | 19.0% | 10.8% | 90.1(0.1) | 78.1(0.2) | 79.9(0.7) | 52.2(0.3) | 85.1(0.1) | 89.2(0.1) | 84.2(0.4) | 79.8(0.1) |

Table 2: Zero-shot results of GPT-Base. "w/o FT" means that we evaluate the pre-trained model without fine-tuning. Our approach achieve similar and even better perplexities than the GPT-Base.

| Method | Saving (FLOPs) | Saving (Walltime) | LAMBADA (w/o FT) | PTB (w/o FT) | WikiText-2 (w/o FT) | WikiText103 (w/o FT) |
|---|---|---|---|---|---|---|
| GPT-Base | 0% | 0% | 54.5 | 146.3 | 49.8 | 50.2 |
| StackBERT | 9.5% | 8.4% | 53.3 | 140.6 | 46.5 | 46.9 |
| bert2BERT | 11.5% | 8.3% | 53.9 | 147.1 | 48.8 | 49.4 |
| LiGO | 14.1% | 6.9% | 54.0 | 139.7 | 50.1 | 50.5 |
| Network Expansion | 15.2% | 12.2% | 54.7 | 143.7 | 50.7 | 51.2 |
| Ours | 24.1% | 16.5% | 53.2 | 142.5 | 47.2 | 47.5 |

**BERT-Base** We compare the results of our method and baselines in Figure 3a and Table 1. The comparison shows that our multi-level framework can save 19.0% FLOPs and 10.8% walltime for

BERT-Base pre-training, while achieving similar down-stream tasks performance as training from scratch. In addition, we find that bert2BERT and KI cannot deliver reduced walltime in our settings.

**GPT-Base**  Figure 3b and Table 2 show the results of GPT-Base model. The results demonstrate that our multi-level framework saves 24.1% FLOPs and 16.5% walltime on GPT-Base pre-training. We observe that our zero-shot results are slightly better than training from scratch.

**DeiT-B**  The results of pre-training DeiT-B are illustrated in Table 3. The multi-level framework saves 27.1% FLOPs and 24.3% walltime in pre-training DeiT-B on ImageNet. Results on downstream tasks prove that our multi-level framework does not impair the model's generalization capability.

Table 3: The transfer learning of DeiT-B. DeiT-B pre-trained with multi-level approach shows a similar level of performance as DeiT-B pre-trained from scratch.

| Method | Saving (FLOPs) | Saving (Walltime) | ImageNet (Top 1 Acc) | CIFAR10 (Acc) | CIFAR100 (Acc) | Flowers (Acc) | Cars (Acc) |
|---|---|---|---|---|---|---|---|
| DeiT-B | 0% | 0% | 81.1% | 99.1% | 90.8% | 97.8% | 92.1% |
| StackBERT | 23.8% | 15.1% | 81.2% | 99.1% | 90.8% | 97.6% | 92.1% |
| bert2BERT | -0.1% | -0.13% | 81.6% | 99.1% | 90.7% | 97.7% | 92.2% |
| LiGO | 25.4% | 12.0% | 81.7% | 99.1% | 90.7% | 97.8% | 92.1% |
| Network Expansion | 25.0% | 22.5% | 81.5% | 99.1% | 90.7% | 97.8% | 92.1% |
| Ours | 27.1% | 24.3% | 81.5% | 99.1% | 90.8% | 97.6% | 92.1% |

**More Levels on BERT-Large**  In our current setting, the number of parameters is reduced by eight-fold in one coalescing operation. Although our framework supports an arbitrary number of levels, we observe that there is little gain on three or more levels in pre-training BERT-Base because the level 3 model has only 1.72M parameters, which is too small to learn from the large dataset. Therefore, we pre-train BERT-Large with two and three levels to evaluate the effectiveness of the number of levels. Figure 3c and Table 4 show the results of the multi-level framework with two and three levels on BERT-Large. When compared with BERT-Base, our approach achieves a higher reduction in FLOPs and walltime on BERT-Large with 2 levels. On the other hand, an increased number of levels results in greater computational savings, indicating our method's potential.

Table 4: Downstream tasks performance of BERT-Large with more levels. Results of BERT-Large indicate that more levels does not lead to performance deterioration while saving more training cost.

| Level | Saving (FLOPs) | Saving (Walltime) | SST-2 (Acc) | MNLI (Acc) | MRPC (Acc) | CoLA (Mcc) | QNLI (Acc) | QQP (Acc) | STS-B (Acc) | Avg |
|---|---|---|---|---|---|---|---|---|---|---|
| 1 | 0% | 0% | 89.6 (0.2) | 79.2 (0.2) | 80.3 (0.8) | 53.0 (1.4) | 86.7 (0.4) | 89.8 (0.1) | 85.0 (0.4) | 80.6 (0.2) |
| 2 | 37.4% | 32.9% | 90.9 (0.3) | 79.8 (0.3) | 78.1 (0.4) | 54.8 (0.1) | 86.9 (0.2) | 89.7 (0.2) | 85.2 (0.4) | 80.8 (0.1) |
| 3 | 51.6% | 41.9% | 90.5 (0.3) | 79.8 (0.4) | 82.6 (0.7) | 56.3 (0.4) | 86.9 (0.1) | 89.9 (0.1) | 84.5 (0.3) | 81.5 (0.1) |

# 5  DISCUSSION

The proposed approach does not need to re-compile the setup of parallel computing of LLMs when transiting from a small model to a larger one as the model architecture and the underlying training environment do not change. Compared with the traditional single level training, the only overhead is incurred by resuming training of the larger model. Here we need to load parameters from storage. In our experiment on BERT-Large, resuming can be done within one minute. This overhead has been taken into account in our results of the walltime saving. For LLaMA-65B, the loading process can be finished in less than 5 minutes on SSD. The estimation is detailed in Appendix C. In summary, the overhead of our proposed V-cycle training is negligible.

# 6  CONCLUSION

This paper proposes an efficient multi-level framework for reducing the training time of transformer-based models. Our framework is built on three basic operators, Coalescing, De-coalescing, and Interpolation, which can be combined to build up a V-cycle training process. The multi-level training offers great potential to deliver a reduced computation time by balancing the fast convergence of smaller models and the high expressiveness of large models. Our experimental results justify the effectiveness of the multi-level training on BERT, GPT, and DeiT models. In addition, our extensive experiments show that our methodology has the potential for saving the computational cost of larger models with more levels. In the future work, we will apply the multi-level framework to the pre-training of large scale models with over 100B parameters.

ACKNOWLEDGEMENTS

We would like to thank Wei Chen, Ke Lin, Bowen Zhang, Yuru Zhang and Dr. Jianguang Lou for valuable discussions on early drafts of this paper. We are grateful to the NLP group of Didi Global Inc. We thank all anonymous reviewers for their insightful feedback.

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

## A  COALESCING AND DE-COALESCING FOR TRANSFORMER

In the Transformer architecture, certain constraints are required to ensure alignment between the input and output across the residual connection, normalization layers, and attention layers. Specifically, embeddings and feed-forward networks (FFNs) only need to adhere to Eq. 2.

**Residual Connection**  Consider layer $j$ follows layer $i$ and has a residual connection, denoted as $\boldsymbol{h}_j = f(\boldsymbol{a}_j) + \boldsymbol{h}_i$, where $\boldsymbol{a}_j$ and $\boldsymbol{h}_j$ represent the outputs of layer $j$ before and after applying the activation function $f$, respectively. It is requisite that $\boldsymbol{F}_{out}^j = \boldsymbol{F}_{out}^i$, ensuring the output alignment of layer $j$ with layer $i$.

**Normalization Layer**  Given that the normalization layer following layer $i$ performs an affine transformation on the normalized result, it is essential that the width coalescing matrix $\boldsymbol{F}_{out}^{norm}$ for parameter $\boldsymbol{W}_{norm}$ is equivalent to $\boldsymbol{F}_{out}^i$.

**Attention Layer**  The attention layer employs matrices $\boldsymbol{W}^Q$, $\boldsymbol{W}^K$ and $\boldsymbol{W}^V$ to derive the queries $\boldsymbol{Q}$, keys $\boldsymbol{K}$ and values $\boldsymbol{V}$, respectively. The output of attention layer can be formalized as follows:

$$Attention(\boldsymbol{Q}, \boldsymbol{K}, \boldsymbol{V}) = softmax(\frac{\boldsymbol{Q}\boldsymbol{K}^T}{\sqrt{d_k}})\boldsymbol{V} \tag{14}$$

where the $d_k$ is the dimension of queries and keys. Given that matrices $\boldsymbol{Q}$ and $\boldsymbol{K}$ are subject to matrix multiplication, we need to limit $\boldsymbol{F}_{out}^Q = \boldsymbol{F}_{out}^K$. Furthermore, as required by Eq. 2, it follows that $\boldsymbol{F}_{out}^Q = \boldsymbol{F}_{out}^K = \boldsymbol{F}_{in}^V$. In the case of multi-head attention layer, these requirements must be satisfied for each head.

## B  WHY NOT INCREASE MODEL SIZE MONOTONICALLY

In this work, we do not increase the model size monotonically as proposed in the previous literature, e.g. LiGOWang et al. (2023). The reason is that the model enlarged from a smaller one in the existing works will have low-rank weight matrices and thus limit the model's capability. The more times the model is mapped, the more severe the limitation becomes. Figure 4 show the training curves of two GPT-Large models. They are trained by following 1) GPT-Small $\rightarrow$ GPT-Base $\rightarrow$ GPT-Large and 2) GPT-Base $\rightarrow$ GPT-Large. The smaller model is mapped to the larger one using LiGO. For fairness, we enlarge GPT-Bases with a close level of performance in 1) and 2) and use the same training settings for the training of GPT-Large. As shown in the figure, the model mapped twice converges significantly slower than the model mapped only once. It confirms

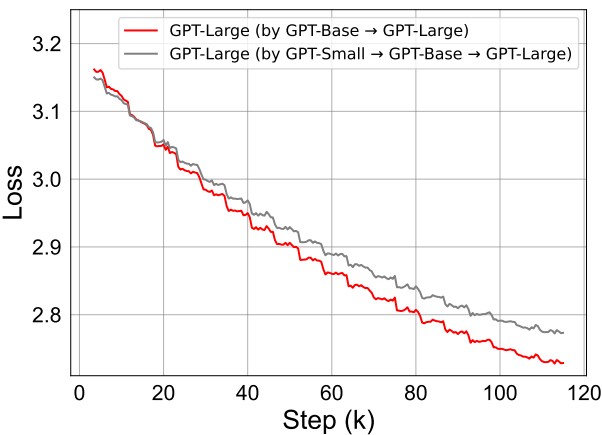

Figure 4: Pre-training GPT-Large mapped once and twice with LiGO. Results show that GPT-Large mapped twice converges significantly slower than GPT-Large mapped once. It confirms that it is not beneficial to monotonically increase the model size as proposed in previous literature.

that it is not beneficial to repeatedly enlarge the model in a monotonic fashion as suggested in the literature. The V-cycle training proposed in this work proves to be advantageous. In Figure 3c, we show that the V-cycle training process with a three-level mapping converges faster than the model with two-level mapping. Therefore, we interpolate the de-coalesced model into the original one instead of directly training the enlarged model.

## C   Deployment Overhead

After interpolating the de-coalesced model into the original one, we need to resume training of the larger model with updated parameters. Since the model architecture and underlying training environment does not change, there is no need to recompile the model, i.e., the setup of parallel computing do not need to be configured again from scratch. The time spent on this step mainly comes from retrieving parameters from storage. Since we re-init the optimizer's parameters when training the interpolated model, we only need to load model parameters from the storage. When pre-training LLMs with fp16, $2\Phi$ bytes are required to save a model with $\Phi$ parameters. For the LLaMA-65B, the amount is around 130GB. A typical HDD will deliver a read speed of 80-160MB/s, and a standard SSD can read data at a rate of 500 to 550MB/s. Therefore, we can load the model parameters within half an hour from HDD and five minutes from SSD. As the training of LLaMA-65B takes approximately 21 days, the resuming cost is negligible.

## D   Effect of Hyper-parameters

Table 5: Effect of Hyper-parameters in our algorithm. Unlisted values are taken to be identical to those of the BERT-Base. $E_a$ is the training steps before coalescing. $E_{small}$ is the training steps for smaller models. The de-coalesced model would be interpolated into the original model with ratio $\alpha$. A full training cycle of BERT-Base is 300K steps. L6-H6 indicates that the model has 6 layers and 6 attention heads, and the same applies to the others.

| | $E_a$ (Steps) | $E_{small}$ (Steps) | $\alpha$ | Coalesced Model Size | Saving (FLOPs) | Saving (Walltime) |
|---|---|---|---|---|---|---|
| BERT-Base | 10K | 150K | 0.5 | L6-H6 | 19.0% | 10.8% |
| (A) | 50K | | | | 9.2% | 1.0% |
| | 100K | | | | 0.8% | -7.4% |
| (B) | | 50K | | | 14.5% | 11.8% |
| | | 100K | | | 17.7% | 6.6% |
| | | 200K | | | 17.8% | 4.4% |
| | | 300K | | | 12.1% | 0.1% |
| (C) | | | 0.05 | | -2.3% | -10.4% |
| | | | 0.25 | | 18.1% | 9.8% |
| | | | 0.75 | | 13.0% | 4.7% |
| | | | 1.0 | | -15.8% | -18.5% |
| (D) | | | | L4-H4 | 12.1% | 5.7% |
| | | | | L8-H8 | 13.4% | 7.1% |
| | | | | L10-H10 | 7.4% | -2.6% |

**Effect of $E_a$**   We demonstrate the effect of hyper-parameter $E_a$ in Table 5 rows (A). Results show that a small $E_a$ is enough. An excessively large $E_a$ would reduce the effect of acceleration because at this time the large model has passed the stage of coarse-grained learning, and the small model no longer plays a role in fast convergence.

**Effect of $E_{small}$**   To study the robustness of the proposed framework, we change the value of the hyper-parameter $E_{small}$ from 50K to 300K. The results in Table 5 rows (B) demonstrate that $E_{small}$ is robust over one half the full cycle. There is no need for a large $E_{small}$ because the smaller models have already passed the fast convergence stage and do not facilitate further acceleration of the training of larger models.

**Effect of $\alpha$**   To investigate the effect of the interpolation operator, we tune the hyper-parameter $\alpha$ from 0.05 to 1.0 for BERT-Base pre-training. Results in Table 5 rows (C) indicate that there is no saving when interpolation operation is removed i.e., $\alpha = 1.0$. The importance of the Interpolation operator for the framework is thus justified. In addition, too small an $\alpha$ cannot efficiently transfer the knowledge from a small model to the large model and thus gives a negative saving ratio. It is also observed that a higher $\alpha$ saves less computational cost because the capability of network is limited.

**Effect of Coalesced Model Size** We demonstrate the effect of the coalesced model size in Table 5 rows (D). Generally, the performance of the pre-trained coalesced model improves as its parameter size increases, but this comes with a corresponding rise in computational cost. The results in Table 5 show that the BERT model with 6 layers and 6 heads offers an optimal balance between the performance and the computational cost, and thus result in the best FLOPs saving.

## E   DETAILS OF COALESCING MATRICES

In Section 4, under the assumption of $d_{in}^{k+1} = d_{in}^k/2, d_{out}^{k+1} = d_{out}^k/2$, we define the width coalescing matrix $\boldsymbol{F}_{w,out}^{k+1,l}$ and the depth coalescing matrix $\boldsymbol{R}^{k+1}$ as follows:

$$\boldsymbol{F}_{w,out}^{k+1,l} = \boldsymbol{H} \otimes \boldsymbol{I} = \begin{bmatrix} 0.5 & \cdots & 0 \\ \vdots & \ddots & \vdots \\ 0 & \cdots & 0.5 \\ 0.5 & \cdots & 0 \\ \vdots & \ddots & \vdots \\ 0 & \cdots & 0.5 \end{bmatrix} \otimes \boldsymbol{I} \tag{15}$$

$$\boldsymbol{R}^{k+1} = \begin{bmatrix} 0.5 & \cdots & 0 \\ 0.5 & \cdots & 0 \\ \vdots & \ddots & \vdots \\ 0 & \cdots & 0.5 \\ 0 & \cdots & 0.5 \end{bmatrix} \tag{16}$$

To demonstrate how the width coalescing matrix coalesces parameters in a transformer model, we decompose the matrix $\boldsymbol{F}$ into matrix $\boldsymbol{H}$ and identity matrix $\boldsymbol{I}$ in Eq. 15. The dimension of the identity matrix $\boldsymbol{I}$ is 64, i.e. the size of attention head in transformer. Thus the matrix $\boldsymbol{H} \in \mathbb{R}^{12,6}$ reveal the way we merge the attention heads. For example, The $\boldsymbol{H}$ we used in Eq. 15 will merge the $i$ and $i+6$ ($i = 1, 2, ..., 6$) attention heads pair by pair.

For simplicity, we denote the width coalescing matrix in Eq. 15 as $\boldsymbol{F}_{out,stack}^{k+1,l}$, and shown the depth coalescing matrix in Eq. 16 with $\boldsymbol{R}_{adj}^{k+1}$. There are different choices for coalescing matrices. For example, we could define the width coalescing matrix $\boldsymbol{F}_{out,adj}^{k+1,l}$ as Eq. 17. And we could define $\boldsymbol{R}_{stack}^{k+1}$ as the inverse operator of stacking layers (Gong et al., 2019).

$$\boldsymbol{F}_{out,adj}^{k+1,l} = \begin{bmatrix} 0.5 & \cdots & 0 \\ 0.5 & \cdots & 0 \\ \vdots & \ddots & \vdots \\ 0 & \cdots & 0.5 \\ 0 & \cdots & 0.5 \end{bmatrix} \otimes \boldsymbol{I} \tag{17}$$

$$\boldsymbol{R}_{stack}^{k+1} = \begin{bmatrix} 0.5 & \cdots & 0 \\ \vdots & \ddots & \vdots \\ 0 & \cdots & 0.5 \\ 0.5 & \cdots & 0 \\ \vdots & \ddots & \vdots \\ 0 & \cdots & 0.5 \end{bmatrix} \otimes \boldsymbol{I} \tag{18}$$

We conduct experiments to verify the effect of coalescing matrices. The comparison reveals minimal differences in FLOPs savings ($< 0.3\%$) among various matrices, which means that the choice of coalescing matrices may not be particularly crucial in our framework.

## F    EFFECT OF COALESCING OPERATION

The coalescing operation establishes a crucial link between the large and small models. To evaluate the effect of the coalescing operation, we remove the coalescing operation in our algorithm, i.e. randomly initialize the small model within the v-cycle. The comparative result, presented in Figure 5a, suffers from an 8.3% drop in FLOPs saving when the small model is randomly initialized. The dramatic drop demonstrates the necessity of the coalescing operation.

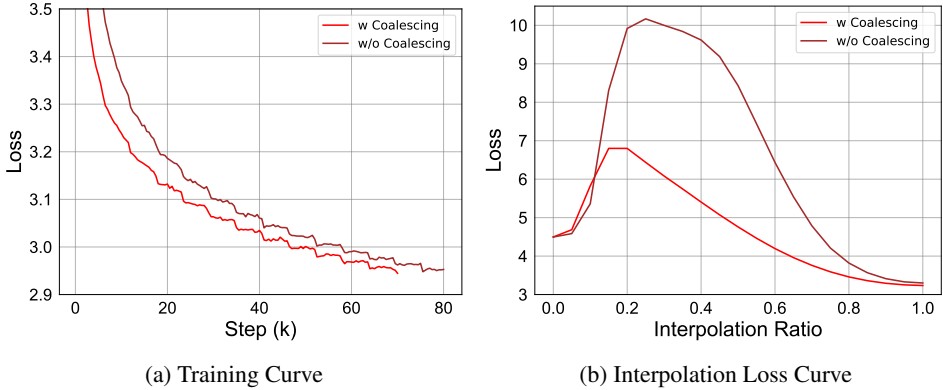

(a) Training Curve    (b) Interpolation Loss Curve

Figure 5: Effect of the Coalescing Operation. In Figure (b), the model corresponds to the GPT-Base before coalescing when the interpolation ratio is set to zero. Conversely, when this ratio is at one, the model becomes equivalent to the de-coalesced model, with or without the application of the coalescing operation.

We examined the interpolation loss curve between the large model prior to coalescing and the de-coalesced model (with or without coalescing). By interpolating models across various alpha values and assessing their validation loss, we could chart a linear path in the optimization spaceGoodfellow & Vinyals (2015). The results, displayed in Figure 5b, reveal lower losses along the interpolation path for the de-coalesced model with coalescing operation. This finding underscores the tighter correlation between the de-coalesced and original models when coalescing is applied.

## G    CONTINUE TRAINING THE DE-COALESCED MODEL

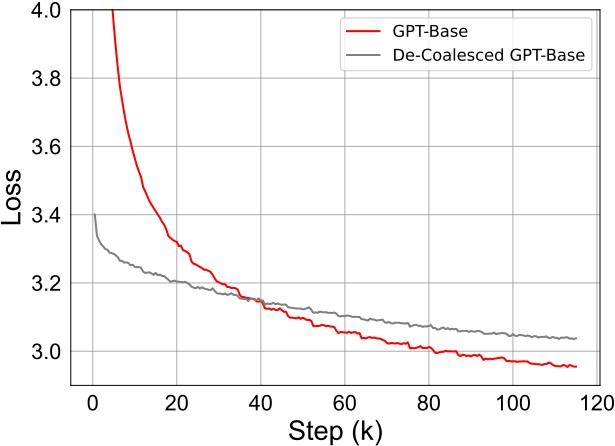

Figure 6: Training Curves of the GPT-Base training from scratch and the De-Coalesced GPT-Base.

To delve deeper into the importance of the interpolation operation, let's consider a straightforward example highlighting the issue of symmetric neurons following width de-coalescing. Consider a two

layers feedforward neural network with an activation function $f$, represented as $f(X\boldsymbol{W}_1 + \boldsymbol{b}_1)\boldsymbol{W}_2 + \boldsymbol{b}_2$. After de-coalescing the network with $\boldsymbol{T}_{out} = [\boldsymbol{I}, \boldsymbol{I}], \boldsymbol{T}_{in} = [\boldsymbol{I}/2, \boldsymbol{I}/2]^T$ in width dimension, the de-coalesced network can be formulated as Eq. 19. For simplicity, let's refer to the de-coalesced model as $f(X\boldsymbol{W}_1' + \boldsymbol{b}_1')\boldsymbol{W}_2' + \boldsymbol{b}_2'$. In this model, the number of first layer's neurons/columns doubles, and the count of second layer's rows increases correspondingly to accommodate the doubled output from the first layer. However, the column count in the second layer remains constant to preserve the output quantity, for example, the number of classes.

$$f(X [\boldsymbol{W}_1, \boldsymbol{W}_1] + [\boldsymbol{b}_1, \boldsymbol{b}_1]) \begin{bmatrix} \boldsymbol{W}_2/2 \\ \boldsymbol{W}_2/2 \end{bmatrix} + \boldsymbol{b}_2 \tag{19}$$

The output of the de-coalesced network is identical to the original. Assuming $n$ neurons/columns in $W_1$. it's straightforward to deduce that $\frac{\partial L}{\partial W_1'[:,1:n]} = \frac{\partial L}{\partial W_1'[:,n+1:2n]}, \frac{\partial L}{\partial b_1'[1:n]} = \frac{\partial L}{\partial b_1'[n+1:2n]}, \frac{\partial L}{\partial W_2'[1:n,:]} = \frac{\partial L}{\partial W_2'[n+1:2n,:]}$. With identical gradients, in the layer 1, the first $n$ neurons will always mirror the subsequent $n$ neurons, implying that the network's learning capability doesn't proportionally increase with the number of parameters.

To illustrate this issue and underscore the interpolation operation's necessity, we continued training the de-coalesced GPT-Base model and have depicted the training curve in Figure 6. The empirical results reveal that due to the limited learning ability caused by symmetric neurons, the convergence performance of the de-coalesced model is significantly inferior to that of the model training from scratch.

## H    RESULTS ON DEIT-S

Table 6: Transfer learning performance of DeiT-S. DeiT-S pre-trained with multi-level approach shows similar performance as DeiT-S pre-trained from scratch. The complexity and redundancy in DeiT-S is less than DeiT-B. Therefore, the FLOPs and walltime saving is less on DeiT-S.

| Method | Saving (FLOPs) | Saving (Walltime) | ImageNet (Top 1 Acc) | CIFAR10 (Acc) | CIFAR100 (Acc) | Flowers (Acc) | Cars (Acc) |
|--------|----------------|-------------------|----------------------|---------------|----------------|---------------|------------|
| DeiT-S | 0% | 0% | 80.1% | 98.9% | 89.8% | 95.4% | 92.2% |
| Ours | 12.2% | 6.5% | 80.1% | 98.9% | 89.3% | 95.4% | 92.3% |

## I    IMPLEMENTATION DETAILS OF OPERATORS

---

**Algorithm 2:** Coalescing Operation

**Input** : A large transformer with $L_1$ layers, hidden dimension of $E_1$, vocabulary size $T$. Denote the embedding weight as $\boldsymbol{W}^{(emb)} \in \mathbb{R}^{T \times E_1}$, parameters of attention layers as $\boldsymbol{W}_l^Q, \boldsymbol{W}_l^K, \boldsymbol{W}_l^V, \boldsymbol{W}_l^O \in \mathbb{R}^{E_1 \times E_1}$, $\boldsymbol{b}_l^Q, \boldsymbol{b}_l^K, \boldsymbol{b}_l^V, \boldsymbol{b}_l^O \in \mathbb{R}^{E_1}$, parameters of FFN layers as $\boldsymbol{W}_l^{(fc1)} \in \mathbb{R}^{E_1 \times 4E_1}, \boldsymbol{W}_l^{(fc2)} \in \mathbb{R}^{4E_1 \times E_1}, \boldsymbol{b}^{(fc1)l} \in \mathbb{R}^{4E_1}, \boldsymbol{b}_l^{(fc2)} \in \mathbb{R}^{E_1}$, parameters of layernorm layer as $\boldsymbol{W}_l^{(ln1)}, \boldsymbol{W}_l^{(ln2)} \in \mathbb{R}^{E_1}, \boldsymbol{b}_l^{(ln1)}, \boldsymbol{b}_l^{(ln2)} \in \mathbb{R}^{E_1}$. Width coalescing matrix $\boldsymbol{F}_{out}^{(emb)}, \boldsymbol{F}_{out}^{(QK)}, \boldsymbol{F}_{out}^{(V)}, \boldsymbol{F}_{out}^{(fc1)} \in \mathbb{R}^{E_1 \times E_2}$. Depth coalescing matrix $\boldsymbol{R} \in \mathbb{R}^{L_1 \times L_2}$.

**Output:** A small transformer with $L_2$ layers, hidden dimension of $E_2$. Utilize an upper right stroke to denote the parameters of the small transformer.

**1** // Preparation

**2** $\boldsymbol{F}_{in}^{(emb)} = \boldsymbol{F}_{out}^{(emb)T} \mathrm{diag}(1/\mathrm{sum}_{\mathrm{col}}(\boldsymbol{F}_{out}^{(emb)} \boldsymbol{F}_{out}^{(emb)T}))$

**3** $\boldsymbol{F}_{in}^{(QK)} = \boldsymbol{F}_{out}^{(QK)T} \mathrm{diag}(1/\mathrm{sum}_{\mathrm{col}}(\boldsymbol{F}_{out}^{(QK)} \boldsymbol{F}_{out}^{(QK)T}))$

**4** $\boldsymbol{F}_{in}^{(V)} = \boldsymbol{F}_{out}^{(V)T} \mathrm{diag}(1/\mathrm{sum}_{\mathrm{col}}(\boldsymbol{F}_{out}^{(V)} \boldsymbol{F}_{out}^{(V)T}))$

**5** $\boldsymbol{F}_{in}^{(fc1)} = \boldsymbol{F}_{out}^{(fc1)T} \mathrm{diag}(1/\mathrm{sum}_{\mathrm{col}}(\boldsymbol{F}_{out}^{(fc1)} \boldsymbol{F}_{out}^{(fc1)T}))$

**6** // Width Coalescing

**7** $\boldsymbol{W}^{(emb)} \leftarrow \boldsymbol{W}^{(emb)} \boldsymbol{F}_{out}^{(emb)}$

**8** **for** $l = 1 \rightarrow L_1$ **do**

**9** $\quad \boldsymbol{W}_l^Q \leftarrow \boldsymbol{F}_{in}^{(emb)} \boldsymbol{W}_l^Q \boldsymbol{F}_{out}^{(QK)}$

**10** $\quad \boldsymbol{W}_l^K \leftarrow \boldsymbol{F}_{in}^{(emb)} \boldsymbol{W}_l^K \boldsymbol{F}_{out}^{(QK)}$

**11** $\quad \boldsymbol{W}_l^V \leftarrow \boldsymbol{F}_{in}^{(QK)} \boldsymbol{W}_l^V \boldsymbol{F}_{out}^{(V)}$

**12** $\quad \boldsymbol{W}_l^O \leftarrow \boldsymbol{F}_{in}^{(V)} \boldsymbol{W}_l^O \boldsymbol{F}_{out}^{(emb)}$

**13** $\quad \boldsymbol{b}_l^Q \leftarrow \boldsymbol{b}_l^Q \boldsymbol{F}_{out}^{(QK)}$

**14** $\quad$ // The bias vector consistently multiplies with the width coalescing matrix.

**15** $\quad$ // For the sake of brevity, we will omit the bias vector in following codes.

**16** $\quad \boldsymbol{W}_l^{(ln1)} \leftarrow \boldsymbol{W}_l^{(ln1)} \boldsymbol{F}_{out}^{(emb)}$

**17** $\quad \boldsymbol{W}_l^{(fc1)} \leftarrow \boldsymbol{F}_{in}^{(emb)} \boldsymbol{W}_l^{(fc1)} \boldsymbol{F}_{out}^{(fc1)}$

**18** $\quad \boldsymbol{W}_l^{(fc2)} \leftarrow \boldsymbol{F}_{in}^{(fc1)} \boldsymbol{W}_l^{(fc2)} \boldsymbol{F}_{out}^{(emb)}$

**19** $\quad \boldsymbol{W}_l^{(ln2)} \leftarrow \boldsymbol{W}_l^{(ln2)} \boldsymbol{F}_{out}^{(emb)}$

**20** **end**

**21** // Depth Coalescing

**22** **for** $l = 1 \rightarrow L_2$ **do**

**23** $\quad \boldsymbol{W}_l'^Q \leftarrow \sum_{i=1}^{L_1} \boldsymbol{W}_i^Q \boldsymbol{R}_{i,l}$

**24** $\quad \boldsymbol{W}_l'^K \leftarrow \sum_{i=1}^{L_1} \boldsymbol{W}_i^K \boldsymbol{R}_{i,l}$

**25** $\quad \boldsymbol{W}_l'^V \leftarrow \sum_{i=1}^{L_1} \boldsymbol{W}_i^V \boldsymbol{R}_{i,l}$

**26** $\quad \boldsymbol{W}_l'^O \leftarrow \sum_{i=1}^{L_1} \boldsymbol{W}_i^O \boldsymbol{R}_{i,l}$

**27** $\quad \boldsymbol{W}_l'^{(ln1)} \leftarrow \sum_{i=1}^{L_1} \boldsymbol{W}_i^{(ln1)} \boldsymbol{R}_{i,l}$

**28** $\quad \boldsymbol{W}_l'^{(fc1)} \leftarrow \sum_{i=1}^{L_1} \boldsymbol{W}_i^{(fc1)} \boldsymbol{R}_{i,l}$

**29** $\quad \boldsymbol{W}_l'^{(fc2)} \leftarrow \sum_{i=1}^{L_1} \boldsymbol{W}_i^{(fc2)} \boldsymbol{R}_{i,l}$

**30** $\quad \boldsymbol{W}_l'^{(ln2)} \leftarrow \sum_{i=1}^{L_1} \boldsymbol{W}_i^{(ln2)} \boldsymbol{R}_{i,l}$

**31** **end**

---

---

**Algorithm 3:** De-Coalescing Operation

---

**Input** : A small transformer with $L_2$ layers, hidden dimension of $E_2$, vocabulary size $T$. Denote the embedding weight as $\boldsymbol{W}^{(emb)} \in \mathbb{R}^{T \times E_2}$, parameters of attention layers as $\boldsymbol{W}_l^Q, \boldsymbol{W}_l^K, \boldsymbol{W}_l^V, \boldsymbol{W}_l^O \in \mathbb{R}^{E_2 \times E_2}$, $\boldsymbol{b}_l^Q, \boldsymbol{b}_l^K, \boldsymbol{b}_l^V, \boldsymbol{b}_l^O \in \mathbb{R}^{E_2}$, parameters of FFN layers as $\boldsymbol{W}_l^{(fc1)} \in \mathbb{R}^{E_2 \times 4E_2}$, $\boldsymbol{W}_l^{(fc2)} \in \mathbb{R}^{4E_2 \times E_2}$, $\boldsymbol{b}_l^{(fc1)} \in \mathbb{R}^{4E_2}$, $\boldsymbol{b}_l^{(fc2)} \in \mathbb{R}^{E_2}$, parameters of layernorm layer as $\boldsymbol{W}_l^{(ln1)}, \boldsymbol{W}_l^{(ln2)} \in \mathbb{R}^{E_2}$, $\boldsymbol{b}_l^{(ln1)}, \boldsymbol{b}_l^{(ln2)} \in \mathbb{R}^{E_2}$. Width coalescing matrix $\boldsymbol{F}_{out}^{(emb)}, \boldsymbol{F}_{out}^{(QK)}, \boldsymbol{F}_{out}^{(V)}, \boldsymbol{F}_{out}^{(fc1)} \in \mathbb{R}^{E_1 \times E_2}$. Depth coalescing matrix $\boldsymbol{G} \in \mathbb{R}^{L_1 \times L_2}$.

**Output:** A large transformer with $L_1$ layers, hidden dimension of $E_1$. Utilize an upper right stroke to denote the parameters of the large transformer.

1 // Preparation

2 $\boldsymbol{T}_{out}^{(emb)} = \boldsymbol{F}_{out}^{(emb)T} \text{diag}(1/\text{sum}_{\text{col}}(\boldsymbol{F}_{out}^{(emb)} \boldsymbol{F}_{out}^{(emb)T}))$

3 $\boldsymbol{T}_{out}^{(QK)} = \boldsymbol{F}_{out}^{(QK)T} \text{diag}(1/\text{sum}_{\text{col}}(\boldsymbol{F}_{out}^{(QK)} \boldsymbol{F}_{out}^{(QK)T}))$

4 $\boldsymbol{T}_{out}^{(V)} = \boldsymbol{F}_{out}^{(V)T} \text{diag}(1/\text{sum}_{\text{col}}(\boldsymbol{F}_{out}^{(V)} \boldsymbol{F}_{out}^{(V)T}))$

5 $\boldsymbol{T}_{out}^{(fc1)} = \boldsymbol{F}_{out}^{(fc1)T} \text{diag}(1/\text{sum}_{\text{col}}(\boldsymbol{F}_{out}^{(fc1)} \boldsymbol{F}_{out}^{(fc1)T}))$

6 // Calculate $\boldsymbol{F}_{in}$ as in Algorithm 2.

7 $\boldsymbol{T}_{in}^{(emb)} = \text{diag}(1/\text{sum}_{\text{row}}(\boldsymbol{F}_{in}^{(emb)T} \boldsymbol{F}_{in}^{(emb)})) \boldsymbol{F}_{in}^{(emb)T}$

8 $\boldsymbol{T}_{in}^{(QK)} = \text{diag}(1/\text{sum}_{\text{row}}(\boldsymbol{F}_{in}^{(QK)T} \boldsymbol{F}_{in}^{(QK)})) \boldsymbol{F}_{in}^{(QK)T}$

9 $\boldsymbol{T}_{in}^{(V)} = \text{diag}(1/\text{sum}_{\text{row}}(\boldsymbol{F}_{in}^{(V)T} \boldsymbol{F}_{in}^{(V)})) \boldsymbol{F}_{in}^{(V)T}$

10 $\boldsymbol{T}_{in}^{(fc1)} = \text{diag}(1/\text{sum}_{\text{row}}(\boldsymbol{F}_{in}^{(fc1)T} \boldsymbol{F}_{in}^{(fc1)})) \boldsymbol{F}_{in}^{(fc1)T}$

11 $\boldsymbol{G} = \boldsymbol{R}^T \text{diag}(1/\text{sum}_{\text{col}}(\boldsymbol{R}\boldsymbol{R}^T))$

12 // Width De-Coalescing

13 $\boldsymbol{W}^{(emb)} \leftarrow \boldsymbol{W}^{(emb)} \boldsymbol{T}_{out}^{(emb)}$

14 **for** $l = 1 \rightarrow L_2$ **do**

15 $\quad$ $\boldsymbol{W}_l^Q \leftarrow \boldsymbol{T}_{in}^{(emb)} \boldsymbol{W}_l^Q \boldsymbol{T}_{out}^{(QK)}$

16 $\quad$ $\boldsymbol{W}_l^K \leftarrow \boldsymbol{T}_{in}^{(emb)} \boldsymbol{W}_l^K \boldsymbol{T}_{out}^{(QK)}$

17 $\quad$ $\boldsymbol{W}_l^V \leftarrow \boldsymbol{T}_{in}^{(QK)} \boldsymbol{W}_l^V \boldsymbol{T}_{out}^{(V)}$

18 $\quad$ $\boldsymbol{W}_l^O \leftarrow \boldsymbol{T}_{in}^{(V)} \boldsymbol{W}_l^O \boldsymbol{T}_{out}^{(emb)}$

19 $\quad$ $\boldsymbol{b}_l^Q \leftarrow \boldsymbol{b}_l^Q \boldsymbol{T}_{out}^{(QK)}$

20 $\quad$ // The bias vector consistently multiplies with the width coalescing matrix.

21 $\quad$ // For the sake of brevity, we will omit the bias vector in following codes.

22 $\quad$ $\boldsymbol{W}_l^{(ln1)} \leftarrow \boldsymbol{W}_l^{(ln1)} \boldsymbol{T}_{out}^{(emb)}$

23 $\quad$ $\boldsymbol{W}_l^{(fc1)} \leftarrow \boldsymbol{T}_{in}^{(emb)} \boldsymbol{W}_l^{(fc1)} \boldsymbol{T}_{out}^{(fc1)}$

24 $\quad$ $\boldsymbol{W}_l^{(fc2)} \leftarrow \boldsymbol{T}_{in}^{(fc1)} \boldsymbol{W}_l^{(fc2)} \boldsymbol{T}_{out}^{(emb)}$

25 $\quad$ $\boldsymbol{W}_l^{(ln2)} \leftarrow \boldsymbol{W}_l^{(ln2)} \boldsymbol{T}_{out}^{(emb)}$

26 **end**

27 // Depth De-Coalescing

28 **for** $l = 1 \rightarrow L_1$ **do**

29 $\quad$ $\boldsymbol{W}_l'^Q \leftarrow \sum_{i=1}^{L_2} \boldsymbol{W}_i^Q \boldsymbol{G}_{i,l}$

30 $\quad$ $\boldsymbol{W}_l'^K \leftarrow \sum_{i=1}^{L_2} \boldsymbol{W}_i^K \boldsymbol{G}_{i,l}$

31 $\quad$ $\boldsymbol{W}_l'^V \leftarrow \sum_{i=1}^{L_2} \boldsymbol{W}_i^V \boldsymbol{G}_{i,l}$

32 $\quad$ $\boldsymbol{W}_l'^O \leftarrow \sum_{i=1}^{L_2} \boldsymbol{W}_i^O \boldsymbol{G}_{i,l}$

33 $\quad$ $\boldsymbol{W}_l'^{(ln1)} \leftarrow \sum_{i=1}^{L_2} \boldsymbol{W}_i^{(ln1)} \boldsymbol{G}_{i,l}$

34 $\quad$ $\boldsymbol{W}_l'^{(fc1)} \leftarrow \sum_{i=1}^{L_2} \boldsymbol{W}_i^{(fc1)} \boldsymbol{G}_{i,l}$

35 $\quad$ $\boldsymbol{W}_l'^{(fc2)} \leftarrow \sum_{i=1}^{L_2} \boldsymbol{W}_i^{(fc2)} \boldsymbol{G}_{i,l}$

36 $\quad$ $\boldsymbol{W}_l'^{(ln2)} \leftarrow \sum_{i=1}^{L_2} \boldsymbol{W}_i^{(ln2)} \boldsymbol{G}_{i,l}$

37 **end**

---

---

**Algorithm 4:** Interpolation Operation

**Input** : A large transformer with $L_1$ layers, hidden dimension of $E_1$, vocabulary size $T$. Denote the embedding weight as $\boldsymbol{W}^{(emb)} \in \mathbb{R}^{T \times E_1}$, parameters of attention layers as $\boldsymbol{W}_l^Q, \boldsymbol{W}_l^K, \boldsymbol{W}_l^V, \boldsymbol{W}_l^O \in \mathbb{R}^{E_1 \times E_1}$, $\boldsymbol{b}_l^Q, \boldsymbol{b}_l^K, \boldsymbol{b}_l^V, \boldsymbol{b}_l^O \in \mathbb{R}^{E_1}$, parameters of FFN layers as $\boldsymbol{W}_l^{(fc1)} \in \mathbb{R}^{E_1 \times 4E_1}$, $\boldsymbol{W}_l^{(fc2)} \in \mathbb{R}^{4E_1 \times E_1}$, $\boldsymbol{b}^{(fc1)_l} \in \mathbb{R}^{4E_1}$, $\boldsymbol{b}_l^{(fc2)} \in \mathbb{R}^{E_1}$, parameters of layernorm layer as $\boldsymbol{W}_l^{(ln1)}, \boldsymbol{W}_l^{(ln2)} \in \mathbb{R}^{E_1}$, $\boldsymbol{b}_l^{(ln1)}, \boldsymbol{b}_l^{(ln2)} \in \mathbb{R}^{E_1}$. A de-coalesced transformer with the same size of the large transformer. Denote the parameters of the de-coalesced transformer with a subscript of "de". Interpolation ratio $\alpha$.

**Output:** An interpolated transformer with the same size of the input transformers. The subscript "intp" is used to designate the parameters of this interpolated transformer.

1   $\boldsymbol{W}_{intp}^{(emb)} \leftarrow (1 - \alpha)\boldsymbol{W}^{(emb)} + \alpha\boldsymbol{W}_{de}^{(emb)}$

2   **for** $l = 1 \rightarrow L_1$ **do**

3      $\boldsymbol{W}_{l,intp}^Q \leftarrow (1 - \alpha)\boldsymbol{W}_l^Q + \alpha\boldsymbol{W}_{l,de}^Q$

4      $\boldsymbol{W}_{l,intp}^K \leftarrow (1 - \alpha)\boldsymbol{W}_l^K + \alpha\boldsymbol{W}_{l,de}^K$

5      $\boldsymbol{W}_{l,intp}^V \leftarrow (1 - \alpha)\boldsymbol{W}_l^V + \alpha\boldsymbol{W}_{l,de}^V$

6      $\boldsymbol{W}_{l,intp}^O \leftarrow (1 - \alpha)\boldsymbol{W}_l^O + \alpha\boldsymbol{W}_{l,de}^O$

7      $\boldsymbol{b}_{l,intp}^Q \leftarrow (1 - \alpha)\boldsymbol{b}_l^Q + \alpha\boldsymbol{b}_{l,de}^Q$

8      $\boldsymbol{b}_{l,intp}^K \leftarrow (1 - \alpha)\boldsymbol{b}_l^K + \alpha\boldsymbol{b}_{l,de}^K$

9      $\boldsymbol{b}_{l,intp}^V \leftarrow (1 - \alpha)\boldsymbol{b}_l^V + \alpha\boldsymbol{b}_{l,de}^V$

10      $\boldsymbol{b}_{l,intp}^O \leftarrow (1 - \alpha)\boldsymbol{b}_l^O + \alpha\boldsymbol{b}_{l,de}^O$

11      $\boldsymbol{W}_{l,intp}^{(ln1)} \leftarrow (1 - \alpha)\boldsymbol{W}_l^{(ln1)} + \alpha\boldsymbol{W}_{l,de}^{(ln1)}$

12      $\boldsymbol{b}_{l,intp}^{(ln1)} \leftarrow (1 - \alpha)\boldsymbol{b}_l^{(ln1)} + \alpha\boldsymbol{b}_{l,de}^{(ln1)}$

13      $\boldsymbol{W}_{l,intp}^{(fc1)} \leftarrow (1 - \alpha)\boldsymbol{W}_l^{(fc1)} + \alpha\boldsymbol{W}_{l,de}^{(fc1)}$

14      $\boldsymbol{b}_{l,intp}^{(fc1)} \leftarrow (1 - \alpha)\boldsymbol{b}_l^{(fc1)} + \alpha\boldsymbol{b}_{l,de}^{(fc1)}$

15      $\boldsymbol{W}_{l,intp}^{(fc2)} \leftarrow (1 - \alpha)\boldsymbol{W}_l^{(fc2)} + \alpha\boldsymbol{W}_{l,de}^{(fc2)}$

16      $\boldsymbol{b}_{l,intp}^{(ln1)} \leftarrow (1 - \alpha)\boldsymbol{b}_l^{(ln1)} + \alpha\boldsymbol{b}_{l,de}^{(ln1)}$

17      $\boldsymbol{W}_{l,intp}^{(ln2)} \leftarrow (1 - \alpha)\boldsymbol{W}_l^{(ln2)} + \alpha\boldsymbol{W}_{l,de}^{(ln2)}$

18      $\boldsymbol{b}_{l,intp}^{(ln2)} \leftarrow (1 - \alpha)\boldsymbol{b}_l^{(ln2)} + \alpha\boldsymbol{b}_{l,de}^{(ln2)}$

19   **end**

---

## J   LEARNED TRANSFORMATION

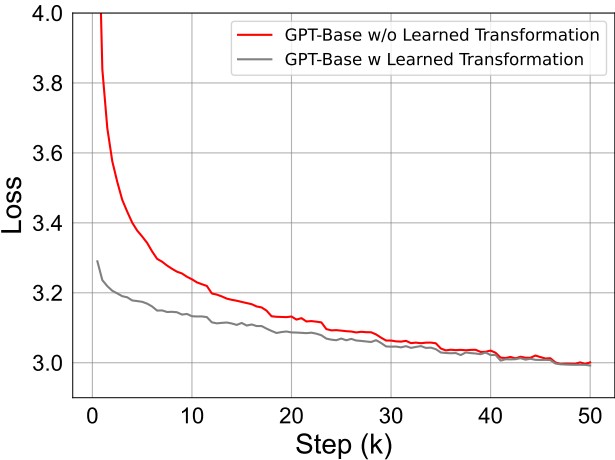

Figure 7: Further Training Curves of the GPT-Base training with or without learned transformation. For learned transformation, we train the de-coalescing matrices as in LiGO to perform a better mapping function, resulting a lower initial loss after interpolated. However, we observe that the model with learned transformation finally converges to the same performance level as the one without learned transformation. The empirical evidence suggests that a direct combination does not lead to advantages.

## K   THE CONNECTION WITH LoRA

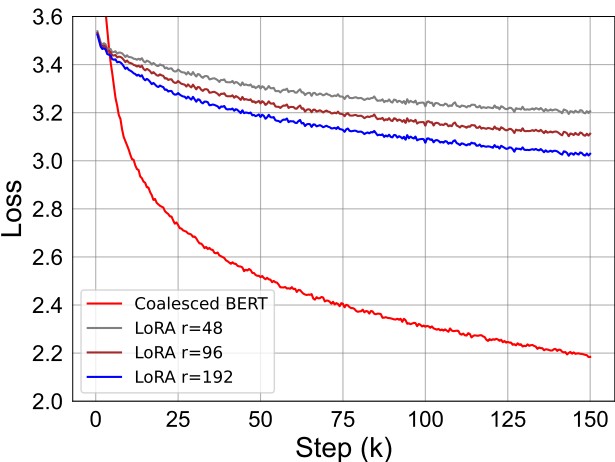

Figure 8: Training Curves of Coalesced BERT and BERT-Base with LoRA: Our findings reveal that the rate of loss decrease in LoRA is substantially slower compared to the coalesced model. Additionally, the FLOPs required for the coalesced model are significantly lower than those for LoRA. Therefore, employing a coalescing operation proves to be far more efficient than using LoRA.

Our framework bears a certain degree of resemblance with LoRA. Notably, when we apply width coalescing alone, the interpolation operator can be redefined as $\boldsymbol{W}_{intp} = \boldsymbol{W}_{large} + \boldsymbol{T}_{in}\boldsymbol{F}_{in}\boldsymbol{W}_{large}\boldsymbol{F}_{out}\boldsymbol{T}_{out}$. Our work focus on updating a segment of the low-rank weights, specifically $\boldsymbol{F}_{in}\boldsymbol{W}_{large}\boldsymbol{F}_{out}$, for both forward and backward propagation. This approach is different from LoRA, which involves using both the original and low-rank weights in the forward propagation. As a result, our approach significantly reduces FLOPs and walltime in the pre-training phase of the

smaller model. In contrast, LoRA does not alter the forward propagation process of the original model and even requires additional computations for the low-rank parameters. While LoRA do not need gradient calculations for weights in the feedforward, layernorm and embedding layers, as well as biases in all layers, the backpropagation process still needs to process gradients from the top layer down by calculating the product of weights in each layer and the gradients of the layer outputs, which is a major contributor to FLOPs in backpropagation. Therefore, the FLOPs saving for LoRA is marginal compared with the coalesced model. Furthermore, our measurements of training time on models starting from scratch and those trained with LoRA revealed similar rates (around 4.0 iterations per second), which further confirms that the computation reduction of BERT with LoRA is marginal. Finally, in Figure 8, comparing the pre-training of BERT-Base with LoRA against Coalesced BERT, we observed that the coalesced model converges much faster than BERT-Base with LoRA. This underscores the importance of focusing on intensive low-rank updates, i.e. the coalescing operation.

