# OpenReview forum: "A Multi-Level Framework for Accelerating Training Transformer Models"
_ICLR.cc/2024/Conference — ICLR 2024 poster_

### Official Review · Reviewer_Csxk · 2023-10-28

**Soundness:** 3 good
**Presentation:** 3 good
**Contribution:** 2 fair
**Rating:** 6
**Confidence:** 5

**Summary:**

The paper introduces a multi-level framework for training large-scale deep learning models like BERT, GPT, and ViT. The framework utilizes operators such as Coalescing, De-coalescing, and Interpolation to exploit inter- and intralayer similarities in feature maps and attentions. It follows a V-cycle training process that progressively adjusts the model size and transfers parameters between levels. Experimental results demonstrate that the proposed framework significantly reduces computational costs  while maintaining performance.

**Strengths:**

1. The paper introduces a novel multi-level framework for training large-scale deep learning models. By leveraging inter- and intralayer similarities, the framework addresses the challenge of high computational costs in training such models. The proposed operators and V-cycle training process provide a unique and effective solution. The V-cycle training process is different to the previous width/depth expansion methods like bert2BERT [1] and network expansion [2].

[1] Chen, Cheng, et al. "bert2BERT: Towards Reusable Pretrained Language Models." Proceedings of the 60th Annual Meeting of the Association for Computational Linguistics (Volume 1: Long Papers). 2022.
[2] Ding, Ning, et al. "Network Expansion for Practical Training Acceleration." Proceedings of the IEEE/CVF Conference on Computer Vision and Pattern Recognition. 2023.

2. The paper supports its claims with extensive experiments conducted on transformer-based language models (BERT, GPT) and a vision model (DeiT). The experimental results demonstrate the effectiveness of the proposed framework, showcasing significant reductions in computational costs while preserving performance.

3. Broad Applicability: The strengths of the paper lie not only in its application to specific models like BERT and GPT but also in its potential applicability to other large-scale deep learning models such as ViT. This suggests that the proposed framework has broader relevance and can contribute to addressing the training cost challenges across various domains and tasks.

**Weaknesses:**

1. The paper lacks in-depth technical explanations about the proposed operators (Coalescing, De-coalescing, and Interpolation) and their implementation. Why the V-cycle training process is better than the previous width/depth expansion methods like bert2BERT and Network Expansion?

2. The paper does not provide a thorough comparison with existing methods or alternative approaches for training acceleration, e.g., Network Expansion [1].

[1] Ding, Ning, et al. "Network Expansion for Practical Training Acceleration." Proceedings of the IEEE/CVF Conference on Computer Vision and Pattern Recognition. 2023.

3. The paper does not extensively discuss the potential trade-offs or limitations introduced by the proposed framework. For example, are there any trade-offs in terms of model accuracy, generalization ability, or robustness? A thorough analysis of these aspects would provide a more comprehensive understanding of the framework's impact on model performance.

**Questions:**

See weaknesses.

---

> ### Author Response · Authors · 2023-11-18
> **Response to Reviewer Csxk**
>
> We thanks reviewer Csxk for acknowledging the broad applicability of our method. We have incorporated all suggestions in the revised version. Below are our responses to the reviewer's questions.
>
> **(1) Detailed explanations about the operators and their implementation**
>
> Thanks for pointing it out. We included comprehensive implementation details of the coalescing, de-coalescing and interpolation operations in Appendix I. Please kindly refer to it for technical explanations.
>
> **(2) Thorough comparison with existing methods**
>
> Thanks for your valuable suggestion! The Network Expansion[1] method, which initializes new filters in CNN by imposing orthogonality and utilizes EMA to expand ViT in depth, is an insightful approach. On BERT, GPT-Base, and DeiT-B, we have added the Network Expansion method and conducted comparative experiments with it. The results of these comparisons are presented in Tables 1, 2, and 3 of the revision. We have also included portions of these tables as follows for your quick reference.
>
> **RTable1: Comparative experiment of Network Expansion in terms of FLOPs Saving**
> | Method            | BERT-Base | GPT-Base | DeiT-B |
> |-------------------|-----------|----------|--------|
> | Network Expansion | 14.8\%    | 15.2\%   | 25.0\% |
> | Ours              | 19.0\%    | 24.1\%   | 27.1\% |
>
> **(3) Trade-offs of the proposed framework**
>
> In our experimental setup, the interpolated model is trained until its validation loss is comparable to that of the baseline model. As a result, hyper-parameters that negatively impact the model's training outcomes tend to show a reduced acceleration ratio, because they require more training steps to match the baseline's performance (the maximal training steps are still the same as baseline). In Appendix D, we maintained a similar validation loss (with differences less than 0.002) across all models to examine the trade-offs of various hyper-parameters in terms of acceleration outcomes. For example, a larger interpolation ratio $\alpha$ might lower the initial loss of the interpolated model, but could also restrict its learning potential. In line with Network Expansion, we determined the optimal post-trade-off $\alpha$ that offers the best balance between acceleration and baseline-level performance. Another key trade-off involves the number of pre-training steps for the small model, which is a balance between enhancing the small model's performance and computational cost. This specific trade-off is also elaborated upon in Appendix D. Lastly, inspired by Network Expansion, we investigated the trade-off related to the size of small model, with the results updated in Appendix D.
>
> **(4) Why the v-cycle training process is better than previous methods**
>
> Thanks for your insightful question. Considering the performance advantages of our approach over bert2BERT and Network Expansion, we note several key distinctions from the previous methods. First, our method expands the transformer model both in width and depth, but bert2BERT and Network Expansion only expand in one dimension, which may limit the potential for performance improvement. Second, bert2BERT adopts a strategy of initializing the expanded parameters in each layer with parameters from higher layers. According to our observations, however, this approach results in a higher initial loss and necessitates more steps for loss reduction, due to the lesser correlation of these inherited parameters with the original ones. Last, Network Expansion utilizes the EMA parameters to enlarge the model depth, which is orthogonal to our method. We would like to incorporate the EMA method into our framework in the future.

---

### Official Review · Reviewer_d5rc · 2023-10-31

**Soundness:** 3 good
**Presentation:** 3 good
**Contribution:** 3 good
**Rating:** 6
**Confidence:** 2

**Summary:**

Transformer-based models perform well in many research areas such as NLP, CV, etc. However, they usually incur exponentially increasing energy costs in the training process. This paper proposes a multi-level framework for training acceleration. The whole working flow is composed of Coalescing, De-coalescing, and Interpolation. More specifically, first, the model is coalesced in both the width and depth direction. Then the large model can be coalesced into a smaller model. Next, to map the parameters back to the original model, the model is depth de-coalesced and then width de-coalesced. Next, after training the smaller model generated by coalescing, it conducts de-coalescing and then merges the coalesced model and de−coalesced model under the control of a hyperparameter. Finally, the merged larger model is trained. The proposed framework is evaluated on both accuracy and speed. The evaluation results show that the framework can keep or even slightly improve the accuracy and reduce the FLOPs and wall time.

**Strengths:**

+ The work proposes a novel method for improving the speed of Transformer-based models.
+ It is carefully written.
+ It offers enough analysis and explanations about the coalescing and de-coalescing details of the Transformer and the reason why this framework is designed in this way.

**Weaknesses:**

- The explanations in section 3 are helpful. However, it would be more helpful if it could include a flow chart or a figure of the structure of the whole framework.
- Algorithm 1 in section 3.4 can help the readers understand the whole flow of the framework but is also kind of sketchy.

**Questions:**

1. How different are the original model and the final model merged by the coalesced model and de−coalesced model? Do they have the same dimension? What are the differences between these two models?
2. What does the number in the brackets represent in Tables 1 and 4?

---

> ### Author Response · Authors · 2023-11-18
> **Response to Reviewer d5rc**
>
> We thanks reviewer d5rc for recognizing the innovation in our method. We have incorporated all the feedback in the revised version. Below are our responses to the reviewer's questions.
>
> **(1) Flow chart to make the structure of the whole framework more clear**
>
> Thanks for your constructive suggestion! Due to the page limit, we were unable to include a separate flow chart. However, we have revised Figure 2 to clarify the overall structure of our framework. Time and model scale arrows have been added at the top and left of the figure to delineate the flow of the framework. Additionally, we relocated the descriptions of operators to positions above their respective arrows for better understanding. We hope these modifications will help readers comprehend the entire framework more effectively.
>
> **(2) Detailed description for Algorithm 1**
>
> Thanks for the recommendation. To provide a clearer understanding, we have added more detailed descriptions (Algorithms 2-4) about the operators in Appendix I. This expansion aims to elucidate Algorithm 1 further and make its processes more understandable.
>
> **(3) How different are the original model and the final model**
>
> Our objective is to accelerate the pre-training of large models. The architecture and parameter size of the final model are identical to those of the original model. While the model size would be reduced during the coalescing stage, we ensure its original dimensions are recovered during the de-coalescing phase.
>
> **(4) The brackets in Table 1 and 4**
>
> The brackets in these tables represent the standard deviation across multiple fine-tuning experiments using different seeds. We have included a description at the top of Table 1 to explain this notation.

---

> > ### Comment · Reviewer_d5rc · 2023-11-22
> >
> > I sincerely appreciate the authors' careful response to my questions. Most of my questions are addressed. However, after reading others' comments, I want to maintain my score.

---

### Official Review · Reviewer_subK · 2023-10-31

**Soundness:** 3 good
**Presentation:** 3 good
**Contribution:** 2 fair
**Rating:** 5
**Confidence:** 4

**Summary:**

This work proposes a multi-level framework for accelerating the training of large-scale deep learning models. This approach is inspired by the observation that training smaller models is more cost-effective and thus the authors propose a solution by generating high-quality intermediate solutions for subsequent larger networks.

Specifically, the authors propose a V-cycle learning process composed of three operations: Coalescing, De-coalescing and Interpolation. The Coalescing operator reduces the model size in terms of width, followed by depth. The De-coalescing operator is the inverse operation of Coalescing, with the de-coalescing matrices defined as the normalized transposition of the coalescing matrices. To address the low-rank issue present in the transformations, the authors also propose the Interpolation operation, which merges the de-coalesced model into the previous one. The authors suggest integrating the three operations into a V-cycle training framework, which learns to coalesce and train small models and then de-coalesce them into bigger models with Interpolation progressively.

The authors also provide experimental results on transformer-based models (BERT, GPT) and a vision model (DeiT), demonstrating significant speed-up (up to >50%) in training while maintaining performance.

**Strengths:**

The idea is clearly presented, and the experimental results appear robust, providing strong support for the conclusions drawn.

**Weaknesses:**

I feel the overall novelty of this paper is a bit limited, as compared with LiGO. I find the major differences lie in two aspects:

 -LiGO learns linear mapping matrices via SGD, while this work intuitively defines the coalescing matrix as $[I, I]^T$, seeking to directly coalesce two neighboring neurons and adjacent layers;
 -As discussed in Appendix B, LiGO gradually learns to increase the model size, whereas this paper introduces V-cycle, a first-coalescing-then-decoalescing learning process equipped with interpolation.

Despite the above, the improvements in FLOPs & Walltime and GLUE over LiGO are marginal (see Table 1). Additionally, more controlled experiments would be beneficial to substantiate the rationale for choosing heuristically defined mapping matrices over learnable parameters. The interpolation operation, which the authors claim mitigates the low-rank issue encountered in LiGO, is reminiscent of well-known PEFT methods like LoRA. More comparisons with this line of research would enhance the persuasiveness and credibility of the proposed method.

Lastly, the authors introduce the multigrid algorithm with a detailed description. However, it seems the proposed framework has little to do with this algorithm.

**Questions:**

What is the significance of the coalescing operation within the overarching framework, and what benefits does it offer compared to initiating the training process with smaller models? In the coalescing step, the compression matrices F_in and F_out in equation 1 and 2, and R in equation 4 are heuristically defined. The recover matrices G in equation 7, and T_in and T_out are also manually defined without further explanation or theoretical basis. I feel the key point of this framework lies in modeling the correlation of parameters between large models and small models. I am not very convinced, from a methodological point of view, why the proposed framework can help to converge faster on the training set D.

Aside from conserving computational resources, what benefits do the interpolating model M_{k} and the de-coalesced model M_{k, de-coalesced} offer compared to continuing training the de-coalesced model M_{k, de-coalesced}?

At the end of the algorithm, the M_{1} model necessitates further training to achieve convergence. I'd like to see the computational overhead of this phase, as well as the comparative experimental results after removing this component.

Most efforts of the experiments are on BERT models. Competitive compared methods, say LiGO, are not included in the results of GPT and DeiT in Table2 & 3.

In Table 4, as the number of Levels increases, the author's method not only saves more computational resources but also further improves the performance of the final model. What is the rationale for this observed enhancement in performance?

---

> ### Author Response · Authors · 2023-11-18
> **Response to Reviewer subK (1/2)**
>
> We are grateful to reviewer subK for the careful inspection and constructive feedback. Our detailed responses to the points raised are as follows.
>
> **(1) Relationship between our algorithm and the multigrid method**
>
> Our algorithm is inspired by the multigrid algorithm proposed for solving large linear systems and bears similarity with it in many aspects. The coalescing and de-coalescing operations in our approach are akin to grid mapping in multigrid. Similarly, the interpolation operation plays the role of the residual correction in multigrid. Furthermore, the overall v-cycle training process corresponds to the v-cycle in multigrid. These parallels reflect a strong conceptual link between our method and the multigrid approach.
>
> **(2) The effect of the coalescing operation**
>
> Thanks for your question about the role of the coalescing operation. The coalescing operation establishes a crucial link between the large and small models. To evaluate the effect of the coalescing operation, we remove the coalescing operation in our algorithm, i.e. randomly initialize the small model within the v-cycle. The comparative result, presented in Appendix F, Figure 5(a), suffers from an 8.3\% drop in FLOPs saving when the small model is randomly initialized. The dramatic drop demonstrates the necessity of the coalescing operation.
>
> Additionally, we examined the interpolation loss curve between the large model prior to coalescing and the de-coalesced model (with or without coalescing). By interpolating models across various alpha values and assessing their validation loss, we could chart a linear path in the optimization space[1]. The results, displayed in Appendix F Figure 5(b), reveal lower losses along the interpolation path for the de-coalesced model with coalescing operation. This finding underscores the tighter correlation between the de-coalesced and original models when coalescing is applied. The detailed discussion of these effects is presented in Appendix F.
>
> **(3) Continue training the de-coalesced model**
>
> Direct training of the de-coalesced model poses practical challenges. The process of de-coalescing involves duplicating attention heads/neurons, resulting in identical gradients for these copied elements, which does not inherently enhance the network's learning capacity. We elucidate this problem mathematically using a simple feedforward neural network example in Appendix G.
>
> We conducted experiments to continuously train the de-coalesced GPT-Base. The training curve, depicted in Appendix G, Figure 6, indicates that the convergence performance of de-coalesced model is markedly inferior compared to the model training from scratch. This empirical result corroborates our theoretical understanding and highlights the limitations of directly training the de-coalesced model.
>
> **(4) The computational overhead of further training $M_{1}$ model and comparative experimental results after removing further training**
>
> In RTable 1, we detail the steps of continuous training that are necessary to achieve comparable performance with the model training from scratch. In RTable 2, we evaluate the zero-shot results of models at different stages, including original GPT-Base before coalescing, the pre-trained coalesced small model, the de-coalesced model, the interpolated model, and post-further-training model. For the de-coalesced and interpolated models, we applied 1K training steps on the training dataset to ensure parameter coherence after mapping. The results in RTable 2 show that:
>
> (a) The Efficiency of De-Coalescing Operation: Our results indicate that the De-Coalescing operation closely mirrors identity mapping, i.e. no decrease in the downstream performance. It implies that the de-coalescing process effectively preserves the learned knowledge from the small model when scaling up to the larger model. However, it is crucial to emphasize that direct continued training of the de-coalesced network, as evidenced above, is not benefitial.
>
> (b) Performance of the Interpolated Model: Compared with the original model before coalescing, the interpolated model demonstrates performance results closely aligned with those of the de-coalesced model. It suggests that the original model effectively assimilates knowledge from the small model and thus accelerates the training process.
>
> **RTable1: Further training steps in different architectures.**
> | Architecture         | Further training steps/epochs | Original model training steps/epochs |
> |---|---|---|
> | BERT-Base            | 214K   | 300K       |
> | BERT-Large (Level-2) | 157K        | 300K       |
> | BERT-Large (Level-3) | 111K        | 300K      |
> | GPT-Base             | 64K     | 115K    |
> | DeiT-B               | 200          | 300     |

---

> ### Author Response · Authors · 2023-11-18
> **Response to Reviewer subK (2/2)**
>
> **RTable2: Zero-shot perplexities of GPT-Base at different stages.**
> | Model   | LAMBADA | PTB    | WikiText-2 | WikiText103 |
> |---|---------|--------|------------|-------------|
> | Original Model (before coalesced) | 199.4   | 1553.5 | 417.7      | 419.9       |
> | Pre-trained Coalesced Model       | 67.6    | 187.5  | 67.8       | 68.1        |
> | De-Coalesced Model                | 65.1    | 174.7  | 66.0       | 66.4        |
> | Interpolated Model                | 86.1    | 230.3  | 97.2       | 97.7        |
> | Further trained Model             | 53.2    | 142.5  | 47.2       | 47.5        |
>
> **(5) Comparison results on GPT and DeiT.**
>
> Thanks for your constructive suggestion. Following your advice, we conducted more comparison experiments on GPT and DeiT during rebuttal. Comparative results are presented in RTable 3 and RTable 4. In addition, we compare with Network Expansion[2] on BERT-Base and update the results in Table 1 of the revised version. These results affirm that our multi-level framework maintains its superiority across various model architectures.
>
> **RTable3: Comparative results on GPT-Base.**
> | Method            | FLOPs Saving  | Walltime Saving     | LAMBADA | PTB   | WikiText-2 | WikiText103 |
> |-------------------|---------|------------|---------|-------|------------|-------------|
> | GPT-Base          | 0\%     | 0\%        | 54.5    | 146.3 | 49.8       | 50.2        |
> | StackBERT         | 9.5\%   | 8.4\%      | 53.3    | 140.6 | 46.5       | 46.9        |
> | bert2BERT         | 11.5\%  | 8.3\%      | 53.9    | 147.1 | 48.8       | 49.4        |
> | LiGO              | 14.1\%  | 6.9\%      | 54.0    | 139.7 | 50.1       | 50.5        |
> | Network Expansion | 15.2\%  | 12.2\%     | 54.7    | 143.7 | 50.7       | 51.2        |
> | Ours              | 24.1\%  | 16.5\%     | 53.2    | 142.5 | 47.2       | 47.5        |
>
> **RTable4: Comparative results on DeiT-B.**
> | Method            | FLOPs Saving  | Walltime Saving     | Imagenet  (Top 1 Acc)  | CIFAR10 | CIFAR100 | Flowers | Cars   |
> |-------------------|---------|------------|-------------|---------|----------|---------|--------|
> | DeiT-B            | 0\%     | 0\%        | 81.1\%      | 99.1\%  | 90.8\%   | 97.8\%  | 92.1\% |
> | StackBERT         | 23.8\%  | 15.1\%     | 81.2\%      | 99.1\%  | 90.8\%   | 97.6\%  | 92.1\% |
> | bert2BERT         | -0.1\%  | -0.13\%    | 81.6\%      | 99.1\%  | 90.7\%   | 97.7\%  | 92.2\% |
> | LiGO              | 25.4\%  | 12.0\%     | 81.7\%      | 99.1\%  | 90.7\%   | 97.8\%  | 92.1\% |
> | Network Expansion | 25.0\%  | 22.5\%     | 81.5\%      | 99.1\%  | 90.7\%   | 97.8\%  | 92.1\% |
> | Ours              | 27.1\%  | 24.3\%     | 81.5\%      | 99.1\%  | 90.8\%   | 97.6\%  | 92.1\% |
>
> **(6) LoRA**
>
> LoRA primarily facilitates the fine-tuning of sizable models using low-rank parameters. Its principal advantage lies in reducing the  memory needed to store optimization states, thereby enabling the fine-tuning of large models on GPUs with a limited memory capacity. However, it's crucial to note that LoRA does not diminish the computational demands associated with forward or backward propagation. This observation suggest that LoRA and our proposed algorithm have divergent objectives. Therefore, it seems infeasible to conduct the comparative experiment for LoRA with a common performance indicator.
>
> **(7) Rationale for the observed enhancement in performance**
>
> Regarding the observed enhancement in Table 4, we find that the enhancement is predominantly the improvements on the MRPC and CoLA datasets. It should be noted that Level-3 does not consistently surpass the baseline across all tests. For instance, its performance is marginally lower than the baseline on the STS-B dataset. Compared with other downstream datasets, the MRPC and CoLA datasets have limited sample sizes, leading to wide fluctuations in finetuning performance. Similar fluctuations are also observed in Table 1 of LiGO. The calculation of the average score does not account for the number of samples or the datasets' reliability and thus result in the average score of level-3 seeming significantly better than baseline actually is. Based on this observation, we could not conclude that level-3 generalizes much better than the baseline.
>
> Lastly, we have implemented learnable mapping matrices in our algorithm. However, our findings suggests that the effect of such an approach is indistinguishable. This is further supported by our experiments on the effects of various coalescing matrices, which indicate that the choice of different coalescing matrices may not be particularly crucial in our set-up.
>
> **References:**
>
> [1] Ian J. Goodfellow, et al. "Qualitatively characterizing neural network optimization problems". ICLR 2015.
>
> [2] Ding, Ning, et al. "Network Expansion for Practical Training Acceleration." Proceedings of the IEEE/CVF Conference on Computer Vision and Pattern Recognition. 2023.

---

> > ### Comment · Reviewer_subK · 2023-11-22
> >
> > I appreciate the comprehensive updates to the content and experimental results, but I still maintain a degree of caution regarding the following two aspects:
> >
> > Coalescing Operation: It appears that the authors have not yet conducted an ablation study on the effects of manually defined matrices, nor have they provided a methodological explanation for choosing this approach over learned transformations.
> >
> > LoRA: My point is that LoRA also involves (de-)coalescing and interpolations from its internal rank perspective, that is, considering the low-rank model as a reduced model. LoRA can indeed reduce FLOPs since it necessitates fewer learnable parameters. While I partially agree that it is orthogonal to your work, it would be intriguing to explore discussions and results of (de-)coalescing from low-rank perspectives.

---

> > > ### Author Response · Authors · 2023-11-23
> > > **Response to Reviewer subK**
> > >
> > > We sincerely appreciate your response and constructive suggestions. Below are our responses to the remaining questions.
> > >
> > > **(1) Learned Transformations**
> > >
> > > We define the coalescing matrices as a binding scheme, deriving from the observed inter- and intra layer similarities, which are a form of inductive prior. We actually already consulted with the author of LiGO regarding the initialization of the mapping matrices in their work, which were not detailed in their paper and the code has not been released due to certain constraints. The author indicated that in LiGO, matrices B and A are initialized uniformly but normalized with a softmax function to mimic a selection matrix, akin to our de-coalescing matrix $T_{out}=[I,I]$. This suggests that in practice, transformations aren't learned from scratch without any inductive priors in LiGO. Although the concept of LiGO is orthogonal to ours and thus it seems that combining both methods could yield further acceleration, our empirical experiments indicate that a learned transformation for de-coalescing matrices doesn't enhance the performance of our framework. This is evidenced in Appendix J, Figure 7, where we show the continuing training of the interpolated model after training the de-coalescing matrices. The results demonstrate that both models eventually converge to similar performance levels with and without learned transformations. The empirical evidence suggests that our approach and LiGO are originated from different perspectives and a direct combination does not lead to advantages.
> > >
> > > **(2) LoRA**
> > >
> > > We agree that our framework bears a certain degree of resemblance with LoRA. Notably, when we apply width coalescing alone, the interpolation operator can be redefined as $W_{intp} = W_{large} + T_{in} F_{in} W_{large} F_{out} T_{out}$. Our work focus on updating a segment of the low-rank weights, specifically $F_{in} W_{large} F_{out}$, for both forward and backward propagation. This approach is different from LoRA, which involves using both the original and low-rank weights in the forward propagation. As a result, our approach significantly reduces FLOPs and walltime in the pre-training phase of the smaller model. In contrast, LoRA does not alter the forward propagation process of the original model and even requires additional computations for the low-rank parameters. While LoRA do not need gradient calculations for weights in the feedforward, layernorm and embedding layers, as well as biases in all layers, the backpropagation process still needs to process gradients from the top layer down by calculating the product of weights in each layer and the gradients of the layer outputs, which is a major contributor to FLOPs in backpropagation. Therefore, the FLOPs saving for LoRA is marginal compared with the coalesced model. Furthermore, our measurements of training time on models starting from scratch and those trained with LoRA revealed similar rates (around 4.0 iterations per second), which further confirms that the computation reduction of BERT with LoRA is marginal. Finally, in experiments comparing the pre-training of BERT-Base with LoRA against Coalesced BERT (Appendix K, Figure 8), we observed that the coalesced model converges much faster than BERT-Base with LoRA. This underscores the importance of focusing on intensive low-rank updates, i.e. the coalescing operation. We have added the discussion about LoRA in Appendix K.

---

### Official Review · Reviewer_2jMP · 2023-10-31

**Soundness:** 3 good
**Presentation:** 3 good
**Contribution:** 3 good
**Rating:** 6
**Confidence:** 2

**Summary:**

The authors propose an efficient multi-level training framework, inspired by the observation of similarities within layers of these models during training. This framework employs a novel approach using three operators: Coalescing, De-coalescing, and Interpolation, to manage model scaling and parameter projection across different model sizes. It introduces a V-cycle training process that alternates between smaller, quickly trained models and larger networks, using the former to provide intermediate solutions for the latter. The interpolation operator is particularly crucial for enhancing convergence by adjusting neuron symmetries after de-coalescing. Experiments show that this framework can reduce computational costs by approximately 20% for BERT/GPT-Base models and up to 51.6% for BERT-Large, without compromising on model performance

**Strengths:**

1. The idea inspired by the multigrid algorithm to accelerate the large model training by coalescing, de-coalescing and interpolation is very clear and promising.
2. Demonstrating the effectiveness of the proposed method is very solid and sound. The interpolation plays an important role in improving the learning ability.
3. The reduction in FLOPs and training time is very significant in NLP transformer models.

**Weaknesses:**

1. Though significant speedup in the NLP transformer, the proposed method has limited improvement in FLOPs and time reduction in the large vision model.
2. It's unclear how to initialize the matrix F. It seems the F can be arbitrary and the initialization of F is not discussed sufficiently.
3. The evaluation result on GPT and DeiT-S lacks a comparison with other works.

**Questions:**

1. Can you explain how the intra- and inter-layer similarity is utilized in the coalescing and de-coalescing procedure?
2. Can you explain why the performance is limited on the vision transformer?

---

> ### Author Response · Authors · 2023-11-18
> **Response to Reviewer 2jMP (1/2)**
>
> We thank Reviewer 2jMP for acknowledging that our idea is promising. We have addressed all the questions and experiments as described below, with corresponding modifications in the revision.
>
> **(1) Initialization of the width coalescing matrix $F$**
>
> Thanks for your question about the initialization of the width coalescing matrix $F$. Here we illustrate the initialization process with a transformer example with 12 layers and 12 heads. Considering to the redundancy in the attention heads as observed in Figure 1, we hope to bind similar attention heads and simultaneously optimize them in a more efficient manner. The width coalescing matrix $F^{k+1, l}_{w, out}=[\mathbf{I}/2, \mathbf{I}/2]^T$ used in our experiments is created by merging the $i$ and $i+6$ ($i=1,2,...,6$) attention heads pair by pair. For the feedforward layer with $n$ neurons, the $i$ and $i+n/2$ neurons is also merged in a paired fashion.
>
> Thanks for the suggestion regarding more discussion on the initialization of F. To providing further comparison, we adopt another width coalescing matrix $F_{adj}$, which merges $i$ and $i+1$ ($i=1, 3,...,11$) attention heads. We give the definition of $F_{adj}$ in Appendix E due to the limitation of OpenReview' to support latex equations. We evaluate the effect of different width and depth coalescing matrices on BERT-Base. Comparative analysis reveals minimal variation in terms of FLOPs savings among various matrices (<0.3\%). We have updated Appendix E with the elaboration on the initialization of width coalescing and the discussion of different width coalescing matrices.
>
> **(2) How intra- and inter layer similarity is utilized in the procedure**
>
> In general, the intra- and inter-layer similarity suggests that it's unnecessary to train similar heads/layers separately. Therefore, we propose the coalescing operation, which binds heads/layers and optimizes them simultaneously. For example, layer 1 and layer 2 would be bound into layer 1' in a smaller model. With the de-coalescing operation, each bound head/layer will then be divided into two identical heads/layers, i.e. layer 1 of the small model is mapped to layer 1 and layer 2 in the de-coalesced model. When interpolated from the de-coalesced, parameters of layer 1 and layer 2 in the original model will be updated in the same way. As a result, the whole procedure can be viewed as updating adjacent heads/layers simultaneously. In this way, we implicitly utilize the intra- and inter-layer similarity to accelerate the training procedure.
>
> **(3) Vision Transformer**
>
> Thank you for raising an interesting question regarding the performance of our framework on vision transformers. In fact, due to limited computing resources before submission, we chose DeiT-S to conduct the vision transformer experiment. DeiT-S has 12 layers but only 6 heads, which means its complexity and redundancy is less than other larger models. Therefore, the savings of FLOPs and walltime are limited on DeiT-S.
>
> After the first submission, we managed to get more computing resources and conducted the vision transformer experiment on DeiT-B, which has 12 layers and 12 heads, i.e. equivalent to the scale of BERT-Base and GPT-Base. Experimental results are presented in the RTable 2. The FLOPs and walltime saving on DeiT-B is 27.1\% and 24.3\%. The results show that our method performs well on vision transformers as long as the parameter size is close to that of other architectures.

---

> ### Author Response · Authors · 2023-11-18
> **Response to Reviewer 2jMP (2/2)**
>
> **(4) Comparison results on GPT and DeiT**
>
> Thanks for pointing out this point. Following your advice, we conducted more comparison experiments on GPT and DeiT during rebuttal. Comparative results are presented in RTable 1 and RTable 2. In addition, we compare with Network Expansion[1] on BERT-Base and update the results in Table 1 of the revised version. These results affirm that our multi-level framework maintains its superiority across various model architectures.
>
> **RTable1: Comparative results on GPT-Base.**
> | Method            | FLOPs Saving | Walltime Saving     | LAMBADA | PTB   | WikiText-2 | WikiText103 |
> |-------------------|---------|------------|---------|-------|------------|-------------|
> | GPT-Base          | 0\%     | 0\%        | 54.5    | 146.3 | 49.8       | 50.2        |
> | StackBERT         | 9.5\%   | 8.4\%      | 53.3    | 140.6 | 46.5       | 46.9        |
> | bert2BERT         | 11.5\%  | 8.3\%      | 53.9    | 147.1 | 48.8       | 49.4        |
> | LiGO              | 14.1\%  | 6.9\%      | 54.0    | 139.7 | 50.1       | 50.5        |
> | Network Expansion | 15.2\%  | 12.2\%     | 54.7    | 143.7 | 50.7       | 51.2        |
> | Ours              | 24.1\%  | 16.5\%     | 53.2    | 142.5 | 47.2       | 47.5        |
>
> **RTable2: Comparative results on DeiT-B.**
> | Method            | FLOPs Saving  | Walltime Saving     | Imagenet  (Top 1 Acc)  | CIFAR10 | CIFAR100 | Flowers | Cars   |
> |-------------------|---------|------------|-------------|---------|----------|---------|--------|
> | DeiT-B            | 0\%     | 0\%        | 81.1\%      | 99.1\%  | 90.8\%   | 97.8\%  | 92.1\% |
> | StackBERT         | 23.8\%  | 15.1\%     | 81.2\%      | 99.1\%  | 90.8\%   | 97.6\%  | 92.1\% |
> | bert2BERT         | -0.1\%  | -0.13\%    | 81.6\%      | 99.1\%  | 90.7\%   | 97.7\%  | 92.2\% |
> | LiGO              | 25.4\%  | 12.0\%     | 81.7\%      | 99.1\%  | 90.7\%   | 97.8\%  | 92.1\% |
> | Network Expansion | 25.0\%  | 22.5\%     | 81.5\%      | 99.1\%  | 90.7\%   | 97.8\%  | 92.1\% |
> | Ours              | 27.1\%  | 24.3\%     | 81.5\%      | 99.1\%  | 90.8\%   | 97.6\%  | 92.1\% |
>
> **References:**
>
> [1] Ding, Ning, et al. "Network Expansion for Practical Training Acceleration." Proceedings of the IEEE/CVF Conference on Computer Vision and Pattern Recognition. 2023.

---

### Author Response · Authors · 2023-11-18
**Summary of Revisions**

We would like to thank all reviewers for their valuable suggestions and constructive comments. Following the reviewers' suggestions, we have revised our manuscript and submitted a new revised version in the "rebuttal revision" field. The revised parts are highlighted in blue color. In the following, we summarize the primary revisions we have made for your convenience to review the revised manuscript.

- Added comparative results on GPT-Base and DeiT-B, in Table 2 and Table 3, as suggested by reviewers 2jMP and subK. We have relocated the DeiT-S experimental results to Appendix H due to the page limit.
- Included a detailed description about the initialization process of coalescing matrices and the impacts, in Appendix E, upon the suggestion of reviewer 2jMP.
- Conducted additional experiments to assess the effect of the coalescing operation, in Appendix F, as advised by reviewer subK.
- Added continuing training results of the de-coalesced model, in Appendix G, as suggested by reviewer subK.
- Expanded Appendix I to include a more comprehensive description of the operators, as suggested by reviewers d5rc and Csxk.
- Clarified the use of brackets in Table 1, as recommended by reviewer d5rc.
- Incorporated references to "Network Expansion"[1] in Section 2.1 (Related Work), Section 4.1 (Experimental Setup), and Tables 1-3, following the recommendation of reviewer Csxk.
- Examined the effect of small model sizes, in Appendix D, as suggested by reviewer Csxk.

---

> ### Author Response · Authors · 2023-11-23
> **Additional Revisions**
>
> We have revised the appendix to better address the questions raised by reviewer subK. Below is a summary of updates:
>
> - Conducted additional experiments for the effect of learned transformations, in Appendix J, as advised by reviewer subK.
> - Explored our method from a low-rank perspective, in Appendix K, as recommended by reviewer subK.

---

### Comment · Area_Chair_nsmW · 2023-11-19
**Please engage in reviewer-author discussion**

Dear reviewers,

The paper got diverging scores. The authors have provided their response to the comments. Could you look through their response and other reviews and engage into the discussion with authors? See if their response changes your assessment of the submission?

Thanks!
AC

---

### Meta-Review · Area_Chair_nsmW · 2023-12-08

**Metareview:**

This work presents a multi-level framework for accelerating the training of large-scale deep learning models.
Specifically, the authors propose a V-cycle learning process which include three operations of coalescing, de-coalescing and interpolation. Experiments show that this framework can reduce computational cost by large margin for BERT and GPT models  without compromising on model performance.

Strengths:
The proposed V-cycle learning process can effectively reduce computational cost in training large models.
Its effectiveness is validated on both transformer-based language models and a vision model (DeiT).

Weaknesses:
Difference between the proposed method and existing methods like LiGO is not well explained.

**Justification For Why Not Higher Score:**

Difference between the proposed method and existing methods like LiGO should be sufficiently analyzed.

**Justification For Why Not Lower Score:**

The proposed V-cycle learning process is novel and effective in reducing computational cost in training large models. Its effectiveness is validated on both large language and vision models.

---

### Decision · Program_Chairs · 2024-01-16

Accept (poster)